# Recovering and monitoring the thickness, density and elastic properties of sea ice from seismic noise recorded in Svalbard

Agathe Serripierri[1], Ludovic Moreau[1], Pierre Boue[1], Jerome Weiss[1], and Philippe Roux[1]

[1]Institut des Sciences de la Terre, Université Grenoble Alpes, Grenoble, France

**Correspondence:** Ludovic Moreau (ludovic.moreau@univ-grenoble-alpes.fr)

**Abstract.** Due to global warming, the decline in the Arctic sea ice has been accelerating over the last four decades, with a rate that was not anticipated by climate models. To improve these models, there is the need to rely on comprehensive field data. Seismic methods are known for their potential to estimate sea-ice thickness and mechanical properties with very good accuracy. However, with the hostile environment and logistical difficulties imposed by the polar regions, seismic studies have remained rare. Due to the rapid technological and methodological progress of the last decade, there has been a recent reconsideration of such approaches. This paper introduces a methodological approach for passive monitoring of both sea-ice thickness and mechanical properties. To demonstrate this concept, we use data from a seismic experiment where an array of 247 geophones was deployed on sea ice in a fjord at Svalbard, between March 1 and 24, 2019. From the continuous recording of the ambient seismic field, the empirical Green's function of the seismic waves guided in the ice layer was recovered via the so-called 'noise correlation function'. Using specific array processing, the multi-modal dispersion curves of the ice layer were calculated from the noise correlation function, and then inverted for the thickness and elastic properties of the sea ice via Bayesian inference. The evolution of sea-ice properties was monitored for 24 days, and values are consistent with the literature, as well as with measurements made directly in the field.

## 1 Introduction

In the alarming context of global warming, modern climatology faces the challenging task of updating climate models for more reliable forecasting. However, these models rely on parameters that are changing at an accelerating rate. One of these parameters is the extent and thickness of the Arctic sea ice, which have been declining faster than expected for the last four decades (Stroeve et al., 2007; Rampal et al., 2011). A worrying example is the change between the forecast from 2012 to 2019: in 2012, a sea-ice-free Arctic was anticipated sometime after 2050 (Stroeve et al., 2012; Steffen et al., 2018), while in 2019, this was modified to as early as 2030 (Screen and Deser, 2019). The evolution of the extent of the sea ice is subject to thermodynamic processes that are affected by important parameters, such as the mechanical properties and thickness of the ice. Hence, continuous and accurate monitoring of these parameters is essential in view of the need to update our climate models.

Currently, given the challenging logistics for accessing the Arctic, satellite remote sensing remains the preferred approach to monitor the thickness of sea ice (Kwok, 2010; Wadhams, 2012). This approach relies on conversion from the sea-ice freeboard distribution into an average thickness, on the assumption that the density of the ice is known. However, this results in high

uncertainties that are due to a number of factors: (i) strong dependence of this approach on models of ocean elevation and the difficulty to correct for fluctuations in the altitude of the ocean surface (Kwok and Cunningham, 2008; Petty et al., 2020); (ii) the presence of snow, which complicates the measurement of the freeboard (Kwok and Cunningham, 2008); and (iii) important temporal and spatial variations in the density of the ice and snow. For example, thin first-year ice (*i.e.*, less than 1-m thick) is typically more porous and contains more brine than thick, multiyear ice. Consequently, depending on the assumptions used to estimate the sea-ice freeboard, and on which density the conversion is made from, estimations of the same sea-ice thickness can vary from 0.2 m to 1.2 m (Ricker et al., 2014; Mu et al., 2018).

While satellite remote sensing solves the problems associated with logistics and allows the monitoring of sea-ice thickness over the whole Arctic Basin, this is at the cost of high uncertainties and poor temporal and spatial resolution. Hence, complementary approaches are needed to improve the monitoring of sea ice. Satellite data can be combined with data acquired in the field, *e.g.*, from upward-looking sonar acquisitions or electromagnetic surveys (Lindsay and Schweiger, 2015; Belter et al., 2020).

The use of seismic methods to study sea ice has been considered for more than 60 years, but due to the hostile environment in polar regions, such studies have remained rare. This is despite their potential for very accurate estimations of the ice thickness, $h$, density, $\rho$, Young's modulus, $E$, and Poisson's ratio, $\nu$ (Anderson, 1958; Hunkins, 1960; Yang and Giellis, 1994; Stein et al., 1998). The main limitation for seismic methods used to be the transport of the seismic stations. However, with the miniaturisation of electronic components and the rapid progress in terms of battery life, it is currently possible to easily transport small autonomous geophones (*e.g.*, less than 1 kg for one geophone) that can record a seismic field for several months. Another limitation was the need for a human presence in the field to proceed with acquisitions using active seismic sources. Seismic methods based on noise interferometry (Shapiro and Campillo, 2004; Sabra et al., 2005) have solved this problem, and seismic acquisitions applied to sea ice are now possible without the need of human intervention, other than for the installation and removal of the geophones (Marsan et al., 2012, 2019; Moreau et al., 2020a, b). Another seismic method that was presented by Romeyn et al. (2021) makes it possible to estimate the thickness of sea ice. This consists of exploiting the propagation of air-coupled flexural waves that are excited by an impulsive seismic source. Romeyn et al. (2021) suggested that these waves might be naturally excited by icequakes produced by natural cracking of the ice, which thus provides a complementary approach to existing passive methods for estimation of ice thickness.

Based on laboratory-scale data, in Moreau et al. (2017), we introduced a methodology where the frequency-wavenumber spectrum of seismic guided waves propagating in an ice layer was used to infer $h$, $E$, and $\nu$ with very high accuracy. In Moreau et al. (2020a), this methodology was successfully applied to field data acquired in March 2019 in the Van Mijen Fjord in Svalbard (Norway), after the frequency-wavenumber spectrum was obtained from noise interferometry via the so-called 'noise correlation function' (NCF). The NCF is obtained by correlating the ambient seismic noise recorded between station pairs of an array of geophones. It can be shown that it converges toward the impulse response of the medium, which for sea ice is a superposition of seismic modes guided through its thickness. Estimations of $h$, $E$, and $\nu$ were obtained from these data, and we concluded that this approach is suitable for long-term monitoring of sea ice. Building on this work, the methodology was improved in view of automatic, more accurate, and more complete monitoring of the ice in the Van Mijen Fjord.

The first improvement concerns the calculation of the NCF. We introduce a method based on beamforming (Rost and Thomas, 2002) for detection and selection of only the time windows where the seismic noise source is aligned with the station pair used, which significantly improves the signal-to-noise ratio (SNR) of the NCF. The second improvement concerns the inversion strategy. In Moreau et al. (2020a), $E$ and $\nu$ were determined from the velocity of the longitudinal and shear horizontal guided waves. The values were then used in a finite element (FE) model for computing synthetic wavefields. However, we noted that a joint inversion of the three parameters $E$, $\nu$, and $h$ is computationally too expensive for forward modeling with a FE model. With these two improvements, we obtained the daily evolution of the thickness and elastic parameters of the sea ice.

In the present paper, we use the analytical model introduced in Stein et al. (1998) for generating synthetics. This approach is computationally very efficient, so Bayesian inference can be used for joint inversion of $E$, $\nu$, and $h$, and also the additional parameter $\rho$ (see section 2.4.2). The evolution of these parameters reveals a constant increase in the ice thickness between March 1 and 24, 2019, while the mechanical parameters remained stable.

## 2 Instruments and methods

Contrary to wave propagation in an infinite and homogeneous domain, propagation in a waveguide (*i.e.*, a thin, bounded domain, such as an ice layer) is subject to multiple reflections at the upper and lower bounds. This causes interference that results in several propagating guided modes that are similar to the Lamb modes that propagate in a stress-free plate (Lamb, 1917). However, the nature of the seismic field in sea ice is modified by the presence of the solid-liquid interface. In the following, we use the terminology introduced in Moreau et al. (2020a) to describe the modes in the wavefield:

- the fundamental quasi-symmetric mode ($QS_0$), which produces mainly longitudinal motion;
- the quasi-Scholte mode ($QS$), which produces mainly flexural motion;
- the fundamental mode ($SH_0$), which produces shear-horizontal motion.

Guided modes are dispersive, and hence seismic signals recorded in sea ice away from the source are distorted. An important property of guided wave propagation is the relationship between the dispersion curves of the guided modes and the mechanical properties and thickness of the ice. The dispersion curves are the representation of the modal propagation in the frequency-wavenumber space, which requires measurement of the wavefield with an array of sensors. In the following section, we only recall the main features of this array, but the reader can refer to Moreau et al. (2020a) for a detailed description.

### 2.1 Seismic array

The experiment was conducted on fast ice in the Van Mijen Fjord, which is located on the southern coast of the island of Spitsbergen, in the Norwegian archipelago of Svalbard (Fig. 1a). More precisely, the seismic array was deployed on Lake Vallunden, which is connected to the fjord by a channel and is surrounded by a moraine. The town of Sveagruva is located not far to the northwest (Fig. 1a).

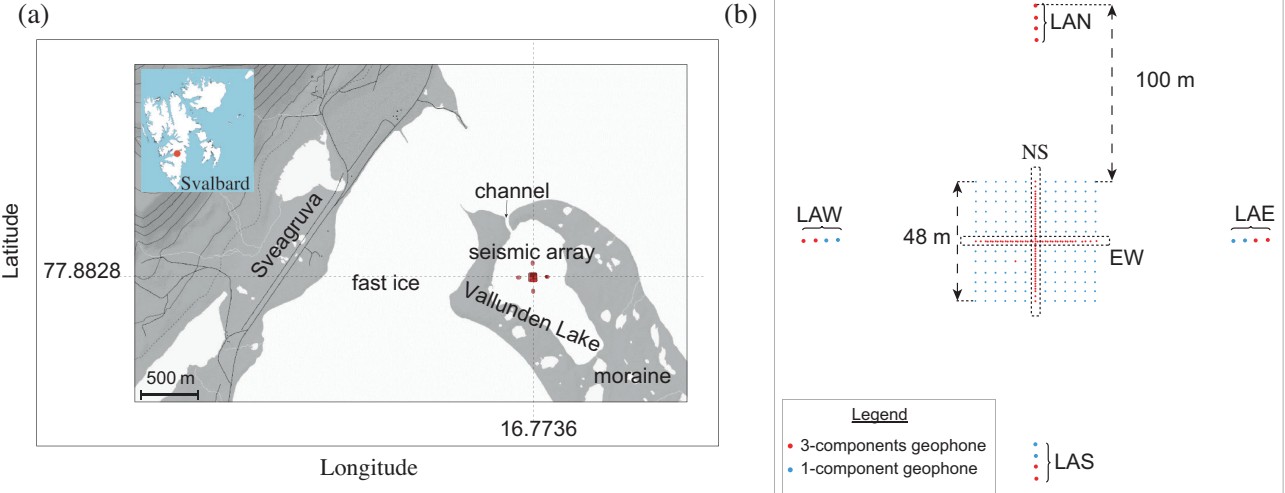

**Figure 1.** (a) Map of the area where the seismic array was installed (red dot) on Lake Vallunden, Svalbard, which is naturally bounded by moraines and connected to Van Mijen Fjord by a small channel. The town of Sveagruva is located about 1 km northwest of the array. (b) Schematic representation of the seismic array. The red dots indicate the three-component (3C) geophones and the blue dots indicate the one-component (1C) geophones. In the main array, the 1C geophones are spaced by 4 m, and the 3C geophones are organized in a dense cross with 1 m spacing oriented along the east-west (EW) and north-south (NS) directions. To the east, north, south, and west there are linear arrays of four geophones (LAE, LAN, LAS, LAW).(a) modified from (Moreau et al., 2020a).

The experiment was meant to deliver a proof of concept for passive seismic monitoring of sea ice, so this place was ideal for several reasons:

- it is close to the facilities of Sveagruva, and easy to access by snowmobile (10 mn drive)

- it is off the main snowmobile travelling routes frequently encountered in the frozen fjords of Svalbard

- despite being surrounded by a moraine, it is subject to tidal forcing and currents, like any other parts with fast ice in Svalbard.

With new processing methods, currently under investigation, that combine deep learning for automatic clustering of the waveforms, with different inversion approaches, it is expected that the same information about sea ice properties can be mon-

itored by deploying only 3-5 stations instead of a dense array, thus significantly easing the logistics associated with transport and deployment of the geophones. The other inversion strategy has recently been tested on drifting pack ice (Moreau et al., 2020b). New deployments on fast ice and on pack ice are being planned to monitor sea ice in the Lincoln sea and in the Nares Strait in 2023 and 2024.

The seismic network is a mix between one-component (1C) and three-component (3C) FairFieldNodal Zland geophones

(Fig. 1b). It consists of a dense array of geophones with 4 m spacing that forms a square. Within this square there is a denser cross of 3C geophones. The first two geophones at the edges of this cross are spaced 2 m apart, and all of the others along the cross have a regular spacing of 1 m. The two lines of geophones that form this dense cross are referred to as the east-west (EW)

line and the north-south (NS) line. To the east, west, north, and south of the array there are extra antennae of four sensors that are referred to as linear array east (LAE), west (LAW), north (LAN), and south (LAS). In the following, LAN and LAS (Fig. 1b) are used as virtual sources for the receivers in the NS line, while LAE and LAE are used as virtual sources for the receivers in the EW line.

The type of geophones used for the experiment has a cylindrical geometry of about 17 cm in height and 12 cm in diameter, mounted on a detachable spike. The geophones were installed directly in the ice without their spike. For accurate positioning, we used taut cords that we attached to each end of the rows and columns of the array, and a decameter. To maximize the coupling, a milling tool was specifically designed to drill the ice at the diameter of the nodes. The snow was removed prior to drilling holes, and geophones were installed in the holes at about half their height. We covered them back with snow to insulate them in view of preserving their battery life. At the time of the deployment, the internal temperature of several nodes was measured, before and after covering them with snow, showing an increase from -21 to -16 °C. Deployment was without particular difficulty, and took about 2.5 days of work for a team of five people, including the time required for their activation. Markers were carefully placed all around the main array and at the position of the four antennae, in order to find them back more easily at the end of the experiment.

The structure of this deployment was designed with a spatial sampling and geometry that allowed the $QS_0$ and $SH_0$ modes to be measured up to about 250 Hz, and the $QS$ mode to be measured up to about 150 Hz Moreau et al. (2020a). Moreover, all of the data used in this work are sampled with a sampling frequency of 500 Hz (Moreau and RESIF, 2019).

## 2.2 Noise correlation function

### 2.2.1 The noise correlation function

For the last 15 years, developments of passive seismological methods have shown that it is possible to extract the medium properties from seismic-noise interferometry without the need for active sources (*e.g.*, sledgehammer, explosions, vibrating truck) or earthquakes. The NCF is obtained by correlation of continuous recordings of seismic noise from a pair of seismometers, and it has been shown to converge toward an estimate of the elastodynamic Green's function between the seismometers (Roux et al., 2005; Campillo and Roux, 2014). This conversion of passive sensors into virtual active sources is very useful in modern seismology for monitoring and tomography purposes (Campillo and Paul, 2003; Shapiro and Campillo, 2004; Sabra et al., 2005). However, theoretical hypotheses that ensure convergence of the NCF toward the Green's function are restrictive, as they require noise sources to be stationary and with an isotropic distribution, and signals to be recorded over an infinitely long time (Lobkis and Weaver, 2001; De Verdière, 2006). In practice, these hypotheses are of course never fulfilled. Nonetheless, the seismic noise is very rich in our dataset: it includes the thousands of icequakes that occur every day, directional anthropogenic noise (*e.g.*, other field experiments in the fjord, human activities near Sveagruva), and seismic noise associated with the wind and water currents. Hence the Green's function can be precisely estimated (Campillo and Paul, 2003; Shapiro and Campillo, 2004) with adequate pre-processing of the data. Recent work provide a catalogue of methods to tackle the challenge of applying passive seismic interferometry to glaciers in the absence of isotropic source distribution (Sergeant et al., 2020).

A usual way of calculating the NCF when sources are not isotropic and/or not stationary consists of truncating the continuous recordings of a station pair into shorter noise segments, and to inter-correlate these segments before averaging the correlations. The segment lengths should be such that seismic sources are as stationary as possible within the segments.

Figures A1 and A2 show the seismic wavefield recorded on 11 March 2019, during the night (figure A1-a) and during the day (figure A2-a) when some fieldwork occurred about 500 m N-E of the main array. The corresponding short-time Fourier transforms are shown in figures A1-b and A2-b, as well as the estimated power spectral densities (figures A1-c and A2-c). These figures show that seismicity can be dominated by noise sources with a very different characteristic time. For example, in presence of human activity the characteristic time is a few minutes (see figure A2-a between 16:00 and 16:35). When there is no human activity, noise sources can be impulsive when icequakes occur (there are also periods of time where many icequakes occur every minute), or they can have a characteristic time of a few minutes (see for example figure A1-a between 2:00 and 2:15). Therefore, it is necessary to correlate noise segments with a length that accounts for the characteristic time of the various noise sources. After some preliminary tests, we concluded that a 5-minutes window is adequate. Moreover, impulsive events, human activity and other noise sources have different levels of energy. To prevent bias due to the dominant noise sources, spectral whitening was applied to the noise segments prior to calculate the correlations. Three sets of correlations were calculated:
- between the recordings from the vertical displacement component
- between the recordings from the north displacement component
- between the recordings from the east displacement component.

In this work, we restrict the study to correlations between i) the 45 stations of the EW line with the 4 stations of the LEA and LAW, and ii) the 45 stations of the NS line with the 4 stations of the LAS and LAN. Of course, by using all stations pairs amongst the 247 stations, it will be possible to obtain 30381 inter-correlations with many different inter-station paths, distances, and directions. With such amount of correlations, it becomes possible to apply tomographic inversions over the full array geometry, and to obtain a 3D+time map of the sea ice properties with unprecedented spatial and temporal resolution. However, this requires a different processing strategy where noise sources have to be selected so that their distribution around the array is isotropic. This is an on-going work that will be the matter of a future paper.

### 2.2.2 Optimization of the noise correlation function using beamforming

The distribution of seismic sources is not isotropic every day, hence we propose to optimize the retrieval of the Green's function by selecting for each day of the recording all of the noise segments for which the energy comes from a direction that is within $\pm 10°$ of the lines of orientation (Fig. 2, red dashed line). The choice of $\pm 10°$ for the direction of the sources is because this aperture is small enough for the emitted wavefield to be localised in the end-fire lobes. These are the areas on either side of the ends of the receiver line in which the phase of the wave correlation function is stationary with respect to the azimuth (Roux et al., 2004). The aperture of these lobes depends on the ratio between the wavelength and the distance between the receivers (Gouédard et al., 2008).

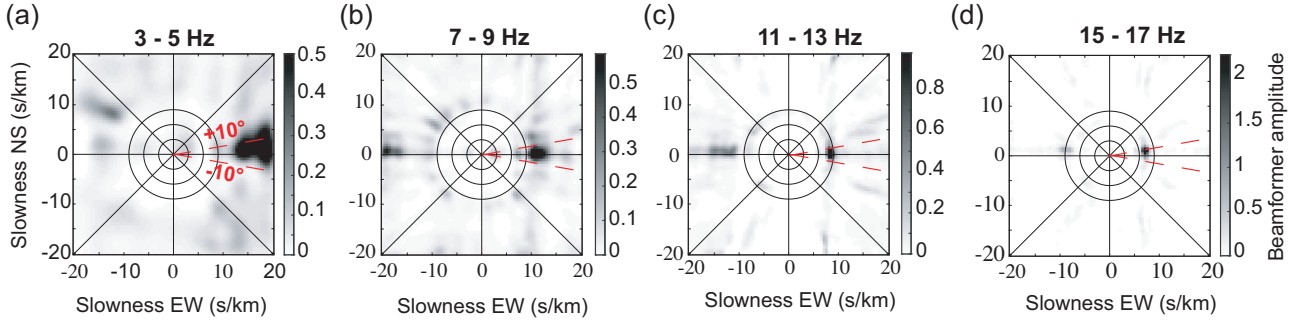

**Figure 2.** Beamforming obtained from the ambient seismic noise recorded on March 9, 2019, over a 5-min window from 3:50 am in four frequency bands: (a) [3-5] Hz; (b) [7-9] Hz; (c) [11-13] Hz; (d) [15-17] Hz. The dominant source is located within $\pm 10°$ (red dashed lines) in the EW sensor line.

To determine the direction of the seismic sources in each noise segment, classical beamforming is applied to the data (Rost and Thomas, 2002). Beamforming was performed using all 1C and 3C stations of the main array that are equally spaced. In some noise segments, seismic sources have different directions for different frequency bands. To prevent the inclusion of these segments in the NCF, we apply the beamforming in several frequency bands, to select only the segments for which the direction of the source is stable independent of the frequency. This processing is based on the spatial coherence between the receivers. Therefore, spatial sampling must satisfy the Nyquist criterion. The 1C geophones have a spatial sampling of 4 m. This sets the minimum wavelength that can be spatially sampled to 8 m, which is that of mode $QS$ at a frequency of 16 Hz. Applying beamforming at frequencies higher than 16 Hz would cause aliasing problems, which prevents dispersion information to be extracted beyond 16 Hz. Hence, we define four frequency bands for selection of the appropriate noise segments: [3-5] Hz, [7-9] Hz, [11-13] Hz, and [15-17] Hz. The direction of the dominant noise source can be identified in each frequency band; *e.g.*, Figure 2 shows the beamforming of a noise segment in all frequency bands. A seismic source from the east is detected, and this segment is therefore selected for calculation of the NCF between the virtual sources of LAE and the receivers of line EW.

Finally, the optimized NCF is obtained by correlation of the selected noise segments and averaging of these correlations. This process is illustrated in Figure 3. Figure3a shows the NCF of all of the 5-min noise segments obtained from the correlation between a virtual source of LAE and the receiver at the centre of the EW line (Figure 1b). The segments selected for the beamforming are shown in color in Figure3a, and the rejected segments (*i.e.*, those showing a superposition of sources that originate from several directions outside the end-fire lobes) are shown in black and white. Indeed, another research team was working in the Vallunden Lake to the north west of our experimental area. This generated many dominant sources from this direction, and impacted upon our seismic array negatively, with a source that was not included in the end fires lobes of our two reception directions, EW and NS. Figure 3b shows the beamforming calculated from all of the 5-min segments (on the left) and that of the selected segments (on the right). Figure 3c shows a comparison between a NCF obtained by averaging the correlations from all of the noise segments (blue waveform) and that obtained by averaging only the selected segments (red waveform). The SNR is significantly improved, especially at early arrival times, where unfavorable noise source directions

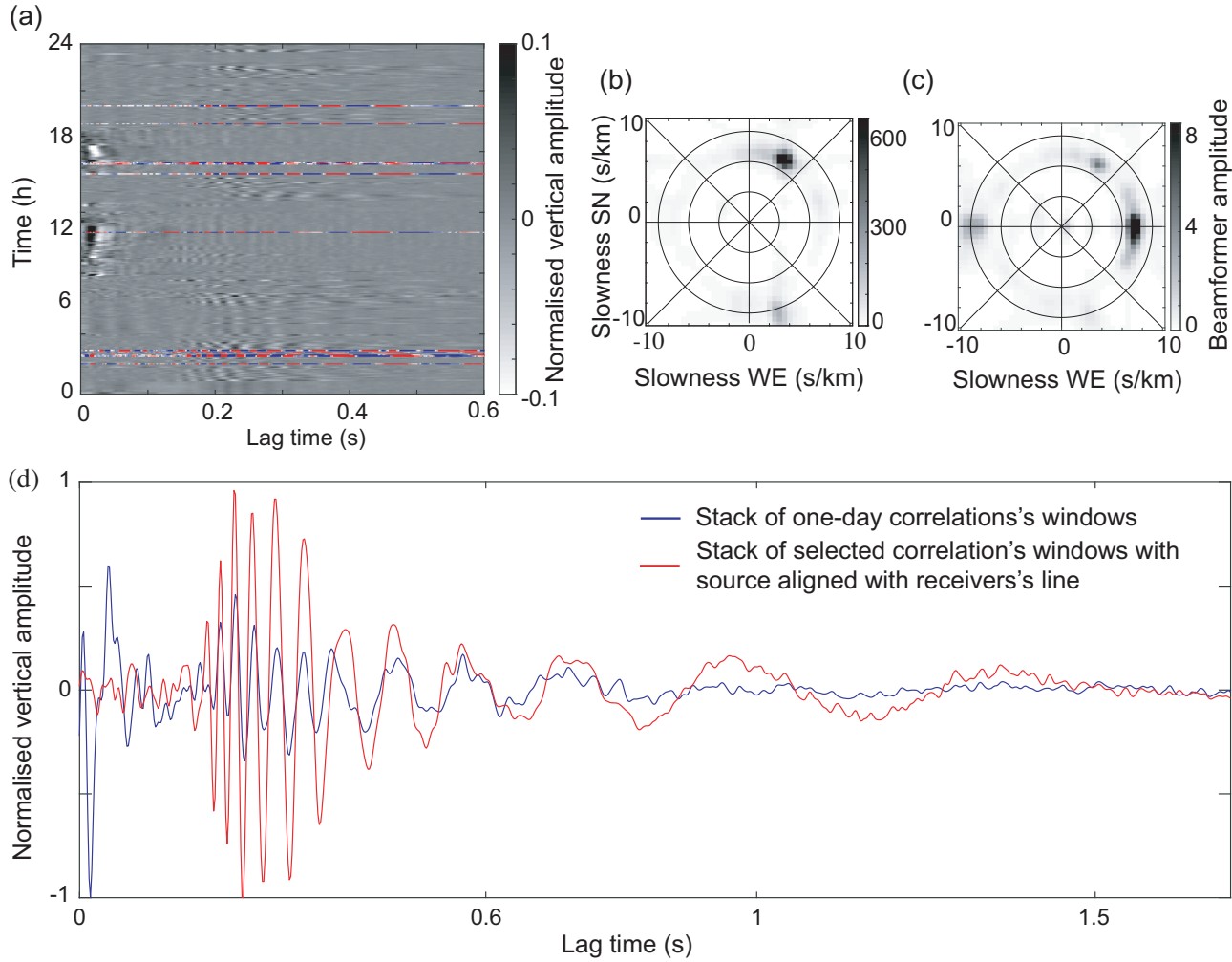

**Figure 3.** (a) Five-minute correlation windows as a function of time between the easternmost station of LAE and the central receiver of the EW sensors line, filtered between 1 Hz and 60 Hz. Correlations shown in color are those with the dominant source aligned with the EW line. The correlations were calculated from the ambient seismic wavefield recorded on March 9, 2019. (b) Beamforming calculated from the full day of ambient seismic noise recorded on March 9, 2019, in the [15-17] Hz frequency band. The noise sources are dominant in the northwest. (c) Beamforming from the summation of the 5-min seismic noise segments selected for an optimal NCF. The dominant noise sources are aligned with the EW line. (d) In blue: stack of the 288 correlations. In red: optimized NCF. The signal-to-noise ratio is significantly enhanced.

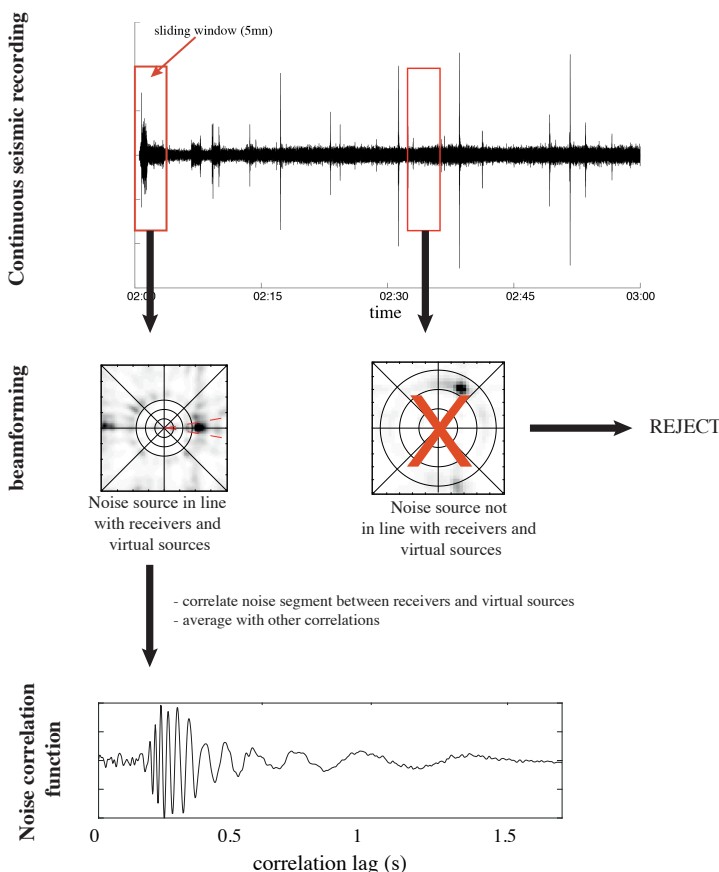

**Figure 4.** Workflow to calculate the noise correlation function from seismic noise.

corrupt the NCF. When calculating the dispersion curves, this introduces a bias in the frequency-wavenumber spectra (see section 2.3), and the modes appear to propagate faster than they actually do. This results in degraded estimates of the sea-ice properties.

The steps to compute the NFC are summarized in figure 4.

## 2.3    Extraction of the frequency-wavenumber spectra from the noise correlation function

To infer the sea-ice properties, the inverse problem is based on a minimization of the difference between the dispersion curves measured *in situ*, and those obtained synthetically. It is therefore necessary to extract the optimal dispersion curve for each mode. To this end, we used the method introduced in Minonzio et al. (2010) in the field of medical ultrasound, and applied

in the context of geophysics on a floating ice layer in Moreau et al. (2017) and Moreau et al. (2020a). As this processing has

already been described several times in the literature, we only recall the main ideas here, and we invite the reader to refer to the above three references for more details.

The linear arrays contain either two or four 3C geophones, which are considered as virtual sources for the calculation of the NCF. The classical way to obtain dispersion curves from a set of NCFs is to apply a Fourier transform to the time and space dimensions, which yields the frequency-wavenumber spectrum. When several virtual sources are available, the spectra can be averaged over the virtual sources.

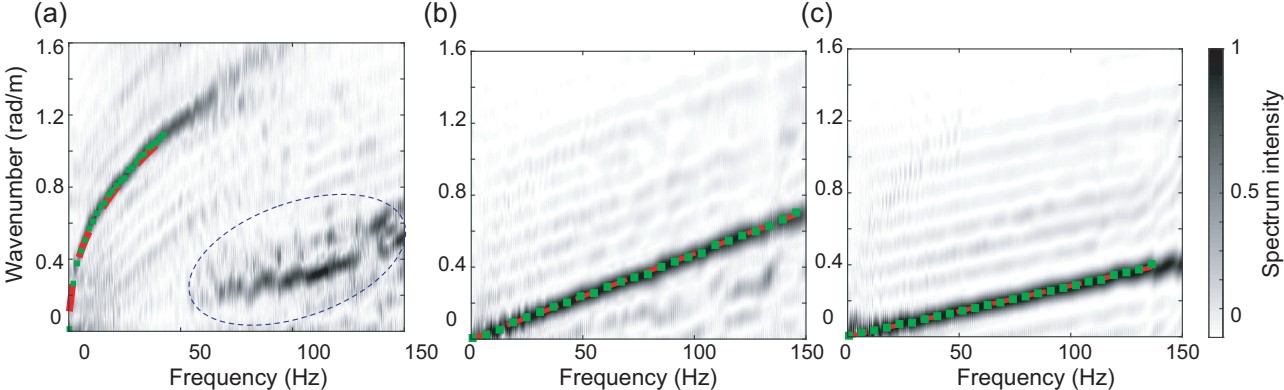

**Figure 5.** Frequency-wavenumber spectra obtained from the noise correlation function calculated on March 9, 2019, with the method described in section 2.2 and the processing of section 2.3, for recordings in the EW direction. (a) Spectrum from the vertical channel, dominated by the $QS$ mode. The zone with a dotted contour highlights a seismic signature with slopes corresponding to a mix between the $SH_0$ and $QS_0$ modes. These modes appear on the vertical channel because the geophones were slightly tilted. (b) Spectrum from the radial channel, dominated by the $QS_0$ mode. (c) Spectrum from the transverse channel, dominated by the $SH_0$ mode. Green dots show the average of the dispersion curves extracted in the EW and WE directions. The red dotted lines show the dispersion curves obtained from Equations (1)-(3), with the sea-ice parameters inferred from Bayesian inference: $E = 4.1$ GPa, $\nu = 0.28$, $\rho = 917$ kg/m$^3$, and $h = 0.60$ m.

A better way of taking advantage of the multiplicity of virtual sources consists of inserting a singular value decomposition to the matrix of the transmit-receive NCF between the Fourier transform on the time dimension and that on the space dimension. This processing consists in the following steps:

1. The matrix of transmit-receive signals has three-dimensions: sources ($M = 2$ or $4$), receivers ($N$), and time. The first step is the application of the Fourier transform to the temporal dimension of this matrix.

2. At each frequency, the resulting Fourier-domain matrix is sliced into 2D transmit-receive matrices. These matrices are then decomposed into singular values. The singular vectors define an orthonormal basis of the space dimensions along the transmitters (left-singular vector) and receivers (right-singular vector). The underlying idea behind this processing step is that the different levels of modal energy are distributed onto the singular vectors, the energy information being contained in the singular values. This allows a heuristic separation of the noise and signal subspaces, in a classical way for singular value-based filters.

3. The last step consists of defining test vectors that are representative of the wave propagation problem. In the present case, we use plane waves of the form $e^{ik_{test}x_n}$, where $k_{test}$ is the wavenumber to be tested, and $x_n$ $(n = 1, 2, ..., N)$ is the coordinate of receiver $n$ along the propagation. Finally, the test vectors are projected onto the singular vectors of the receivers' basis. This leads to a scalar product that is maximized when the wavenumber in the test vector matches that of the waves in the measured wavefield. In practice, this projection step is equivalent to calculating the discrete spatial Fourier transform of each singular vector.

Step 3 is performed at each frequency resulting from step 1.

Once steps 1-3 are performed, the resulting frequency-wavenumber spectrum significantly enhances the identification of the dispersion curves, for two reasons: (i) it is possible to separate signal from noise by applying a threshold to the singular values; and (ii) modal amplitude stands out at all frequencies and for all modes with the same spectral intensity (Fig. 5), despite their different relative amplitude in the wavefield, because singular vectors have a unit norm. The dispersion curves can therefore be identified on a larger bandwidth and with less SNR-related uncertainties than with conventional beamforming techniques (Moreau et al., 2020a). In the present work, they are extracted from the spectrum by identification of all of the frequency-wavenumber couples with a spectral intensity above a threshold. This threshold is set heuristically to 0.2, to achieve the best compromise between the visibility of the propagation modes and that of the seismic noise that pollutes our spectra. Based on this method, four sets of dispersion curves are extracted:

- from E to W with LAE as virtual sources;

- from W to E with LAW as virtual sources;

- from N to S with LAN as virtual sources;

- from S to N with LAS as virtual sources.

Finally, the dispersion curves obtained from propagations in opposite directions are averaged together to reduce uncertainties, in a similar fashion as in (Moreau et al., 2014b). This yields one dispersion curve per mode per day in the EW and NS lines. Next, the dispersion curves are inverted for the sea-ice properties.

## 2.4 Estimation of sea-ice parameters with Bayesian inference

### 2.4.1 Forward model, parameterization of the problem, and cost function

In the following, we use the notation $k_m^{\text{type}}(f)$ to describe the frequency-wavenumber spectrum of a mode $(m = QS, QS_0, SH_0)$ at frequency $f$. Type $=$ "EW" and "NS" refer to wavenumbers obtained from the data in the EW and NS lines, respectively, and Type="syn" refers to synthetic wavenumbers obtained with the forward model. The forward model used here is that introduced by Stein et al. (1998), which describes the asymptotic behavior of the seismic wavefield in a thin ice sheet that is floating on an infinite water column, for low values of the product $f \cdot h$. While the depth of the lake is only about 10 m (Marchenko et al., 2021), this is sufficient for the {ice-water} system not to behave like a bi-layer waveguide, because the first dispersive mode in an acoustic waveguide of thickness 10 m is evanescent under 75 Hz. The wavenumbers are obtained by solving the following

equations:

$$k_{QS_0}(\omega) = \omega \sqrt{\frac{\rho(1-\nu^2)}{E}} \tag{1}$$

$$k_{SH_0}(\omega) = \omega \sqrt{\frac{2\rho(1+\nu)}{E}} \tag{2}$$

$$k_{QS}(\omega)^4 - \frac{h\rho\omega^2}{D} - \frac{\rho_w}{D}\left(\frac{\omega^2}{\sqrt{\left(\frac{k_{QS}(\omega)}{\omega}\nu\right)^2 - \left(\frac{\omega}{c_w}\right)^2}} - g\right) = 0. \tag{3}$$

In these equations, $\omega$ is the angular frequency, $c_w$ and $\rho_w$ are the speed of sound and density of the water, respectively, and $D = \frac{Eh^3}{12(1-\nu^2)}$ is the ice-bending rigidity.

Equation(1) holds as long as the $QS_0$ does not become dispersive, which occurs at $f \cdot h$ values above 500 Hz·m. The sampling frequency of the signals in this study is 500 Hz, so this condition is always satisfied. Equation(2) is always valid, because the fundamental guided shear-horizontal mode, $SH_0$, is not dispersive. Equation(3) is valid when $f \cdot h$ remains less than about 50 Hz·m. The ice in the Van Mijen Fjord had a thickness of less than 0.8 m, hence this Equation(3) can be used at frequencies up to ~65 Hz.

Given a dataset, $\mathbf{d}$, and a set of parameters for the ice layer, $\mathbf{X} = \{E, \nu, \rho, h\}$, a cost function can be defined as the L2 norm between the wavenumbers calculated from the data and those calculated from Equations (1)-(3), such that:

$$f(\mathbf{d}, \mathbf{X}) = \frac{1}{3} \sum_{m=QS, QS0, SH0} \left\| k_m^{syn}(\omega) - k_m^{EW, NS}(\omega) \right\|, \tag{4}$$

where $\|.\|$ refers to the L2 norm. In Moreau et al. (2020a), $E$ and $\nu$ were calculated directly from Equations (1) and (2), assuming the density is known. Then $h$ was estimated based on a grid search and forward modeling with the FE method. FE modeling was used because it allows the inversion to be performed with high frequencies, of up to 100 Hz. The downside is that it is computationally very expensive. In the present paper, instead of a FE model, we preferred to use Equation (3), which is much more efficient, even though it means that we are limited to frequencies under 65 Hz for the $QS$ mode. The benefits are two-fold:

- First, compared to FE modeling, the efficiency of using Equation (3) allows inversions based on Bayesian inference, for $h$, $E$, and $\nu$, and also for $\rho$. The probability density function (PDF) of these parameters can be estimated and used to evaluate the uncertainties of the inversions.

- Secondly, Equation (3) constrains mainly the thickness (to the power of three), but it also slightly constrains $E$, $\nu$, and $\rho$ (to a unit power). Hence, the problem is better constrained when performing a joint inversion of all of the parameters simultaneously.

### 2.4.2 Estimation of the probability density function of the ice parameters with Bayesian inference

Solving the inverse problem consists of finding the parameters that best explain the data, based on the cost function defined in Equation (4). This is a well-posed and well-constrained inverse problem, because there is a one-to-one relationship between the model parameters and the global minimum of the cost function. We proceed with the Markov Chain Monte Carlo (MCMC) algorithm (Metropolis et al., 1953), which provides an ensemble of solutions that fit the data with an acceptable level of likelihood, given the data uncertainty. This ensemble of solutions is represented by the posterior distribution of the model parameters, such that:

$$P(\mathbf{X}|\mathbf{d}) = \frac{P(\mathbf{d}|\mathbf{X})\,P(\mathbf{X})}{P(\mathbf{d})}. \tag{5}$$

where $P(\mathbf{X}|\mathbf{d})$ is the likelihood function, $P(\mathbf{X})$ is the prior distribution, and $P(\mathbf{d})$ is the marginal likelihood function, which is essentially a normalization factor. The posterior distribution expresses the conditional probability of the parameter values based on evidence from measurements, expressed by the likelihood function, and from prior assumptions, expressed by the prior distribution. It is also an estimate of the parameter PDF.

In the present problem, it is assumed that measurement errors are not correlated and are random, and thus that they can be modeled by a zero-mean Gaussian likelihood function with variance $\sigma^2$:

$$P(\mathbf{d}|\mathbf{X}) = \exp\left(-\frac{(f(\mathbf{d},\mathbf{X}))^2}{2\sigma^2}\right), \tag{6}$$

where $\sigma^2$ is the variance associated with the measurement errors. This is a typical likelihood function as used in many data-fitting problems (Tarantola, 2005).

For first year ice, in situ measurements of density range from 840 to 910 kg/m$^3$ for the ice above the waterline, and 900 to 940 kg/m$^3$ for the ice below the waterline (Timco and Frederking, 1996). Poisson's ratio varies between 0.25 and 0.4, Young's modulus between 2 and 6 GPa (Anderson, 1958; Timco and Weeks, 2010). In particular, in the Van Mijen fjord, Romeyn et al. (2021) reported a Young's modulus around 2.5 GPa with a Poisson's ratio of 0.33, and Morozov et al. (2011) reported a Young's modulus around 3 GPa and a Poisson's ratio of 0.3. (Moreau et al., 2014a), reported a Young's modulus around 4 GPa and a Poisson's ratio of 0.32 at Vallunden. The slightly higher value of Young's modulus at Vallunden, in comparison with those directly in the Van Mijen fjord, is likely attributable to the protected physical setting of the study site, and support from the surrounding shoreline at the moraine. Regarding the thickness, ice drillings indicated that it was systematically less than 1 m. Hence, we assume for the prior distributions that the model parameters have equal probability over a range of values that contains the above-referenced values:

- $E$ is between 2 GPa and 6 Gpa;

- $\nu$ is between $0.1$ and $0.5$;

- $\rho$ is between 700 kg/m$^3$ and 1000 kg/m$^3$;

- $h$ is between 0.15 m and 1.15 m.

Determining the optimal value of $\sigma^2$ is essential for appropriate sampling of the PDF. However, $\sigma^2$ is data-dependent. In our approach, we are inverting dispersion curves, so it should account for uncertainties in the dispersion curves, not those in the measurements. Unfortunately, there is no linear relationship between both, because of the dispersion of the QS mode. Depending on the quality of the correlation function, these uncertainties are not the same between all sets of dispersion curves. Hence $\sigma^2$ cannot be set *a priori* for all of the inversions in this study, or else it would be a coarse approximation.

Moreover MCMC methods generally require a burn-in phase before reaching the posterior distribution. To tackle both of these issues, we precede the MCMC algorithm by simulated annealing (SA) global optimization (Moreau et al., 2014a; Gradon et al., 2019) to determine $\sigma^2$ and to improve convergence. Another approach consists of considering the variance as part of the inversion parameters of the MCMC algorithm, in what is know as trans-dimensional MCMC. However the method proposed in this paper has the advantage that it converges much faster, and it is robust for dealing with our dataset.

Simulated annealing is also an MCMC method . However, its aim is to converge on the global optimum rather than to generate samples from the posterior distribution (Kirkpatrick et al., 1983). To this end, $\sigma^2$ in the likelihood function is initialized with a large value, to allow broad sampling of the parameter space. This value is slowly reduced with each iteration, in what is termed the "cooling schedule", and if a sufficiently gradual schedule is chosen, the algorithm will converge to the global optimum. In practice, the cooling schedule can be implemented in a number of ways. We use a standard schedule, $T$, where the variance is reduced from $T_0 = 0.05$ to $T_1 = 0.001$ in an exponential function such that:

$$T(n) = T_0 \left( \frac{T_1}{T_0} \right)^{n/N} \tag{7}$$

The number of iterations, $N$, is set to 20000 for the SA. The SA phase is stopped when the algorithm reaches a number of iterations equal to $N$, or when it has remained stuck at the same position in the search space for more than 200 successive iterations. The MCMC algorithm is then started, with 50000 iterations, from the position in the search space that minimizes the cost function during the SA phase, and with a variance equal to the minimal variance reached, increased by 1% (which translates to a 10% increase of the standard deviation). This increase is to avoid the MCMC algorithm from spending hundreds of iterations (like the SA) at the same position in the search space, which would be a sign of poor calibration.

The variance is linked to measurement errors; *i.e.*, a mix between the influence of the SNR and other random perturbations of the measure, such as variations in the physics of the problem. In the present problem, such perturbations are averaged when going from the time-space to the frequency-wavenumber domains, and thus the variance reflects variations of frequency-wavenumber values around ground truth values. Figure 5a-c indicates that the spread of the frequency-wavenumber couples is very narrow. Calculations of this spread show that it remains of the order of $\pm$ 0.002 rad/m, which corresponds to a maximal variance of about 4e-6 (rad/m)$^2$. This is consistent with the order of variances found with our method, as shown in Figure 6 for all of the days from March 1 to March 24, 2019. It appears stable around 0.5e-6 in the EW direction. In the NS direction, however, it shows a slight decrease. Since the SNR remains stable in the NCF, this decrease appears to be linked to temporal and spatial variations in the physics of the problem, which cannot be accounted for with the forward model. One of these is the influence of the snow layer on the seismic waveforms. In the NS direction, on March 1, the snow cover was thinner near LAS

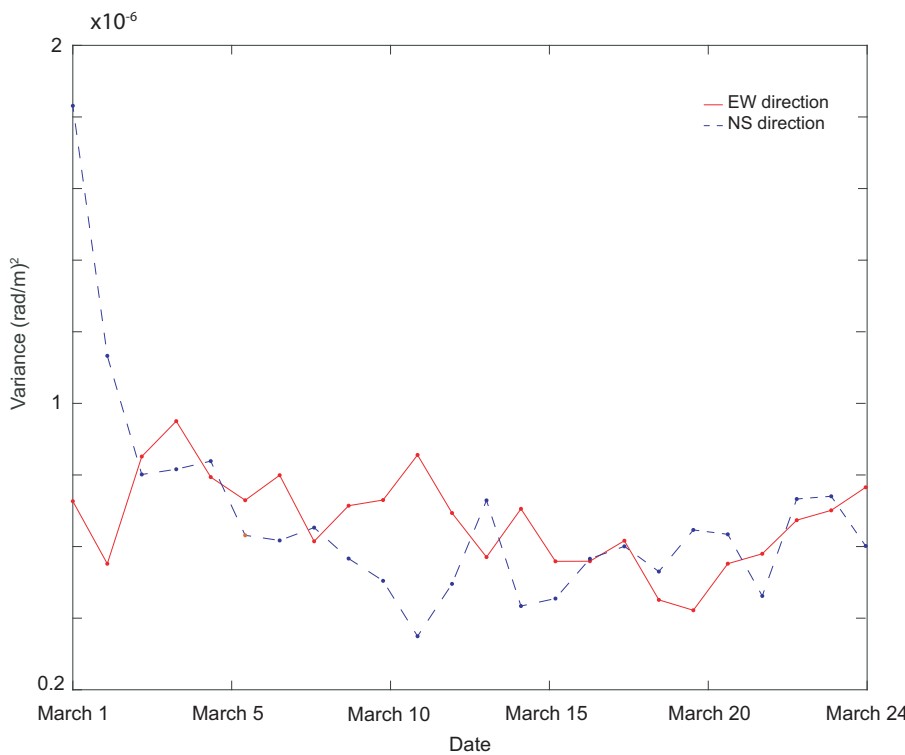

**Figure 6.** Daily evolution of the variance of measurement errors, estimated with the simulated annealing algorithm, as described in section 2.4.2. Variance in the EW (red) and NS (blue) directions. The variance in the EW direction remains stable, at around 0.5e-6. That in the NS direction appears to show a slightly decreasing pattern, which appears to be the consequence of temporal variations of the ice properties in this direction.

(about 10 cm) than it was near LAN (about 40 cm). Such variations were not noted in the EW direction, where the thickness of the snow was more constant. Additional rainfall and snowfalls in March are likely to have influenced the propagation of the seismic waves by modifying the apparent density and Young's modulus of the ice+snow system. This should be investigated in future studies, and forward modeling that accounts for a snow layer would represent significant progress.

After the MCMC algorithm has completed, a PDF is generated for $E$, $h$, $\nu$, and $\rho$. Figure 7 shows an example of PDF obtained on March 9, from line EW. The ice properties are determined by computing the histograms of the parameters, to which we fit with a kernel-type distribution density estimator. The kernel estimation method creates a smooth, continuous curve to represent the probability distribution of the data from the data samples by estimating locally the normal distribution function centered in each sample (the kernel functions). Summing these local smoothing functions for each sample produces

the resulting continuous fit. We use the maximum of this fit to determine the estimate of the parameters. Figure 5 shows the dispersion curves obtained from the estimated parameters; *i.e.*, $h = 0.60$ m, $E = 4.1$ GPa, $\nu = 0.28$, and $\rho = 917$ kg/m$^3$. We observe a statistically consistent fit between the theoretical model and the wavefield data.

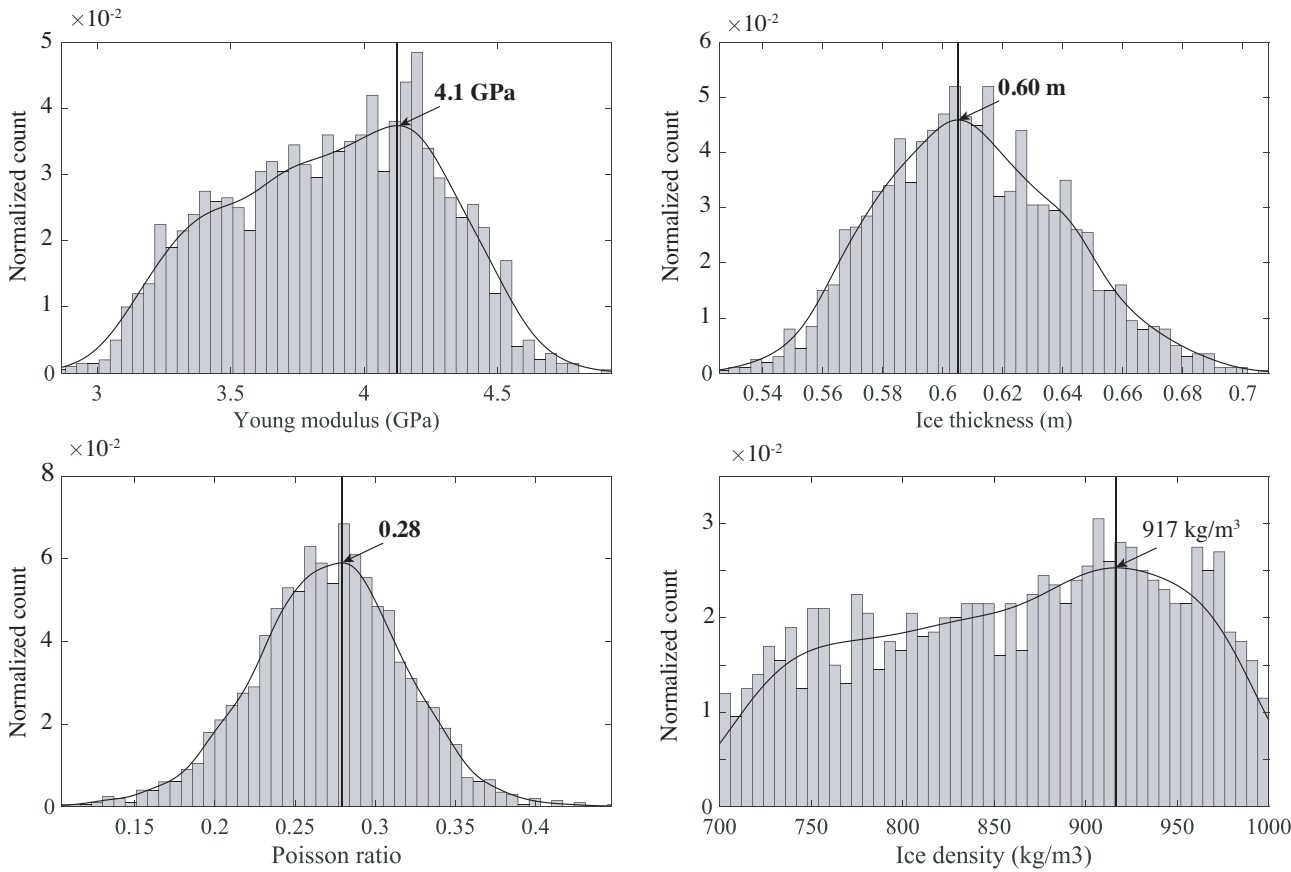

**Figure 7.** Example of the probability density function for $E$, $h$, $\nu$, and $\rho$ obtained after the inversion of the dispersion curves calculated on March 9, 2019, for LEW. The red curve is an estimate of the distribution of the PDF calculated from the kernel density estimator. The blue curve shows the maximum of our distribution. The Young's modulus, the ice thickness, and the Poisson's ratio are satisfactorily constrained by our method. The ice density is less well constrained, with a flatter distribution. The values obtained are nevertheless consistent with the literature.

From the PDF, we find that the thickness, Young's modulus, and Poisson's ratio are satisfactorily constrained, with a standard deviation of 3 cm, 0.4 GPa, and 0.04, respectively. There is clear sharpening of the PDF around the most probable values. The PDF of the density, however, is not as sharp, with a standard deviation of about 80 kg/m3. Interestingly, the covariance of the parameters (see figure B1 in Appendix B) indicates that Poisson's ratio, despite being well-constrained, seems rather uncorrelated from the other parameters. On the other hand, Young's modulus, density and thickness appear to be strongly correlated, despite the density being not very well-constrained.

These observations seem to indicate that density having a flatter PDF reflects the limits of our forward model more than it is an indicator of the limits of the methodology. The model is only sensitive to the effective properties of the {ice+snow} system, because it cannot account for the snow layer (about 40 cm thick, on average), which modifies the effective properties. The weight of the snow layer modifies the density of the {ice+snow} system more than it does its rigidity (Young's modulus) and expansion/contraction (Poisson's ratio). Presumably, a forward model able to account for snow would be a significant improvement, which should constrain the density in a better way. The development of such a forward model is therefore an important follow up of this work. Moreover, the values obtained from all inversions remain constant around $\rho = 915$ kg/m$^3$, which suggests that the actual standard deviation is not as high.

## 3    Results and discussion

Sea ice is a complex material that is made up of solid ice, salt water, solid salts, and gases bubbles. In the Van Mijen Fjord, its growth is mainly controlled by the weather conditions *in situ*, but other factors are involved, such as: marine currents, which are mainly dominated by the tides; the supply of glacial freshwater from the surrounding mountains, which lowers the salinity of the water in the fjord (Høyland, 2009); and warm Atlantic water from the West Spitsbergen Current (Gerland and Hall, 2006). The Van Mijen Fjord ice that we are studying is seasonal first-year ice, which disappears every summer, and it is covered by snow.

The estimated parameters are of major importance for understanding and modeling sea-ice dynamics. Thickness has a key role in coupled modeling (Allard et al., 2018), while the elastic parameters have key roles in recent rheological models (Dansereau et al., 2016). Figure 8 shows the evolution of the four parameters obtained by performing one inversion per day for the NS and EW sensor lines, and over a period from March 1 to 24, 2019. The estimations are consistent with the values reported in the literature for first year ice, which are given in section 2.4.2.

In the field, a snow layer of 10 cm to 40 cm covers the ice. It is likely that this has an influence on our estimations, because guided modes propagate at a lower speed in snow than in ice. Hence, it is possible that some parameters might have been slightly under-estimated. However, is difficult to explicitly constrain the influence of the snow layer on our estimations, and this is left for a dedicated study.

The ice thickness is directly controlled by external environmental factors, such as: air temperature, snow type and depth, wind speed, ocean heat flux, and surface radiation. For the EW direction, we observe a quasi-linear increase in ice thickness from March 1 to 24. The thickness varies from $55 \pm 3$ cm to $66 \pm 3$ cm. For the direction NS, we also observe an almost linear

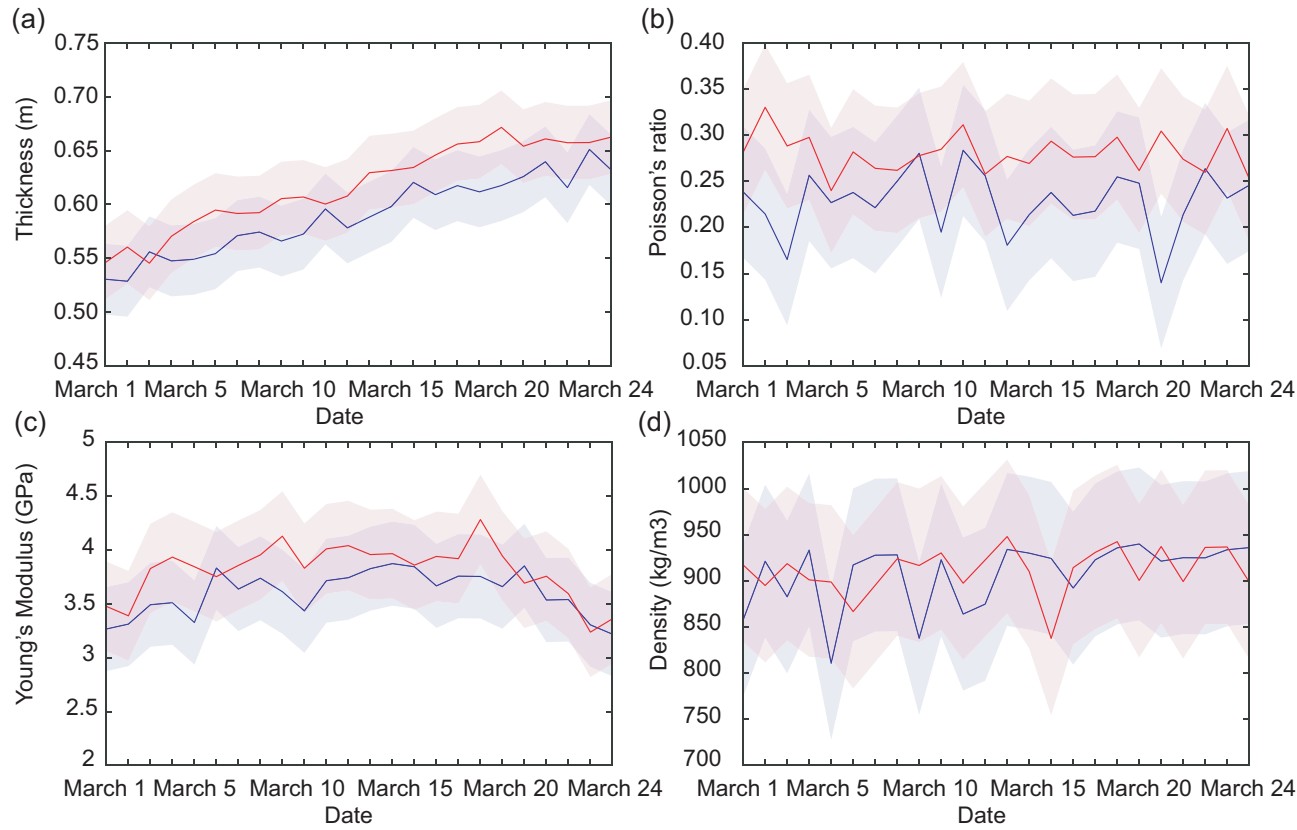

**Figure 8.** Daily evolution of the sea-ice parameters: (a) thickness, (b) Poisson's ratio, (c) Young's modulus, and (d) density, obtained for the lines of sensors in direction NS (blue) and EW (red). The coloured band represents the standard deviation associated with each of the curves. We observe an increase in the ice thickness in both directions, while the elastic parameters remain stable, around values that are consistent with the literature for first-year ice.

increase in ice thickness that ranges from $53 \pm 4$ cm to $63 \pm 4$ cm. These thickness values are consistent with the measurements made during the installation (about 60 cm) and de-installation (about 70 cm) of the array (Moreau et al., 2020a), although they are slightly lower. This difference can be explained by the presence of the snow on the ice. We suspect that the calculated thickness could be under-estimated by 2 cm to 3 cm, but this falls within the uncertainty margins. Note that the inferred ice
thicknesses differ systematically by 2-3 cm in the NS and EW directions. A likely explanation could be related to the elongated geometry of the moraine in the NS direction, which could have a mechanical effect to the ice while it is growing. Also, the channel to the north, where water current is present could have an effect on the ice thickness. However, it is noteworthy that these differences between both directions are not statistically significant, since the uncertainties overlap, hence the differences could also be linked to the quality of the Green's function retrieval.
The Young's modulus depends on several parameters, such as salinity, porosity, and density of the ice. At Lake Vallunden, the ice contained a lot of brine, and was relatively porous and brittle for the first 30 cm; it was also covered with snow. All of

these affected the speed of the guided modes, which propagate at a reduced speed compared to freshwater ice. The Young's modulus remains around $E = 3.7 \pm 0.4$ GPa for both directions.

The Poisson's ratio can be influenced by many factors too, such as loading rate, temperature, grain size, grain structure, direction of loading, and state of microcracking, among others. In view of the short study period, we therefore expect to have a relatively stable Poisson's ratio of around $0.28 \pm 0.05$ for the EW direction and $0.23 \pm 0.05$ for the NS direction.

Knowing the density of the ice and its thickness are crucial parameters for safe ice transport and the opening of new shipping routes. The salinity and porosity of the ice are factors that influence its density. The density of sea ice is not constant through the ice thickness. It tends to be less dense in the freeboard. This is because air bubbles tend to accumulate in the upper part of the ice, while brine tends to accumulate in the lower part of the ice (Høyland, 2009). The density is generally insensitive to temperature changes, which is consistent with our results. The inversions give a value that is very stable, at around $910 \pm 82$ kg.m$^{-3}$ for the EW direction, and $908 \pm 80$ kg.m$^{-3}$ for the NS direction.

From these inversion results, we note that the thickness, Young's modulus, and Poisson's ratio are very well constrained by our method. All of our results appear to be compatible with relatively small thickness and young ice that insulates the heat fluxes rather poorly. The density parameter is a bit less well constrained. The PDF of the density has a more uniform distribution, and the uncertainty is larger than for the other parameters. Nevertheless, the actual uncertainty is probably much smaller, as all of the inversions give a very stable value, of around 910 kg/m$^3$ for the EW direction and 908 kg/m$^3$ for the NS direction. Moreover, the density values obtained are within a range confirmed by the literature (Timco and Frederking, 1996). In this sense, we can choose to perform the same inversions by fixing the density, on the assumption that we can better constrain the other parameters. In this way, we obtain the same results with smaller uncertainties. They are 0.1 GPa for the Young's modulus, $0.04$ for the Poisson's modulus, and 2 cm for the ice thickness. As the results obtained are very close when the density is either fixed or not fixed, it appears scientifically very interesting to also invert the density parameter.

## 4  Conclusions

This paper introduces a methodology for estimating and monitoring the thickness, Young's modulus, density, and Poisson's ratio of sea ice in different directions, using the ambient seismic noise recorded with a seismic array. The methodology consists of extracting from the NCF, and then inverting, the dispersion curves of the guided modes propagating in sea ice. To calculate the daily NCF, we show that selecting the time windows where the dominant seismic source is aligned with the receivers significantly improves the SNR. This strategy also prevents potential directional bias from other dominant noise sources, which would result in higher apparent modal velocity and corrupt the estimation of sea ice parameters. The dispersion curves of the three fundamental guided modes are inverted with MCMC sampling for inferring the probability density function of the sea ice parameters. We obtain satisfactory results that are consistent with the observations and measurements made *in situ*. Thus, we demonstrate that by using this method, it is possible to constrain the thickness, density, and elastic properties of sea ice both precisely and on a daily basis.

This approach could also be useful in various other fields, such as maritime transport safety or oceanography. The daily monitoring of ice thickness and stiffness offered by this method can help to ensure safe movement on sea ice in polar regions where the majority of the population travels by snowmobile. It can also open up possible new routes that were previously closed due to the lack of knowledge of these parameters. In the field of biology, for example, observations suggest that Meso-Zooplankton Biomass Density is closely related to variations in sea-ice properties.

In future works, another way of processing these seismic data is to exploit the thousands of icequakes present in the recordings, which arise from the natural cracking of the ice. These energetic and impulsive sources will allow the reconstruction of the Green's function of the medium in a multitude of directions over time. This will form the basis for a temporally evolving three-dimensional tomographic profile of the sea ice beneath the seismic array, with applications for characterizing its rheology.

**Appendix A: Examples of typical signals recorded during night and day at Vallunden.**

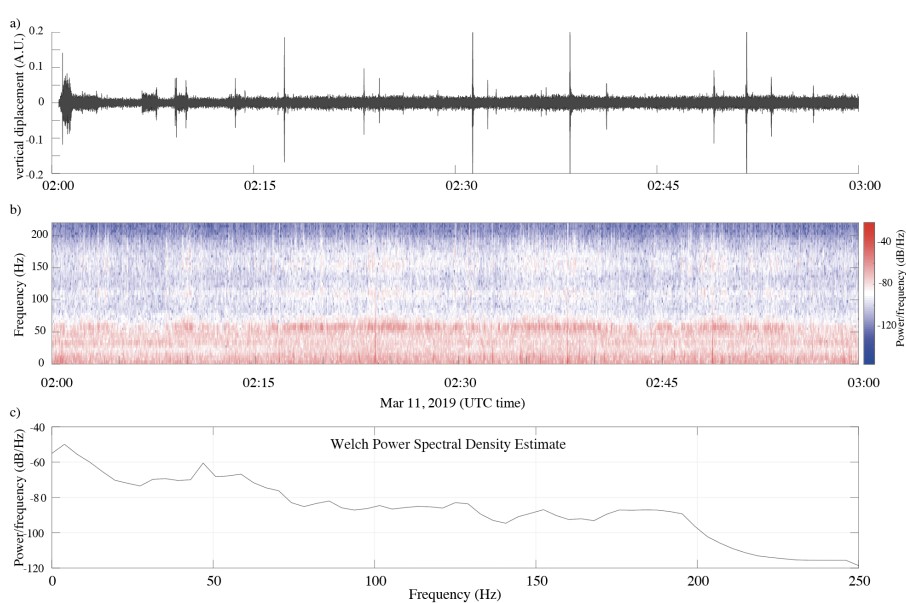

**Figure A1.** a) One hour of seismic wavefield at night, on 11 March 2019, with b) the corresponding short time Fourier transform, and c) Welch Power Spectral density of the recording

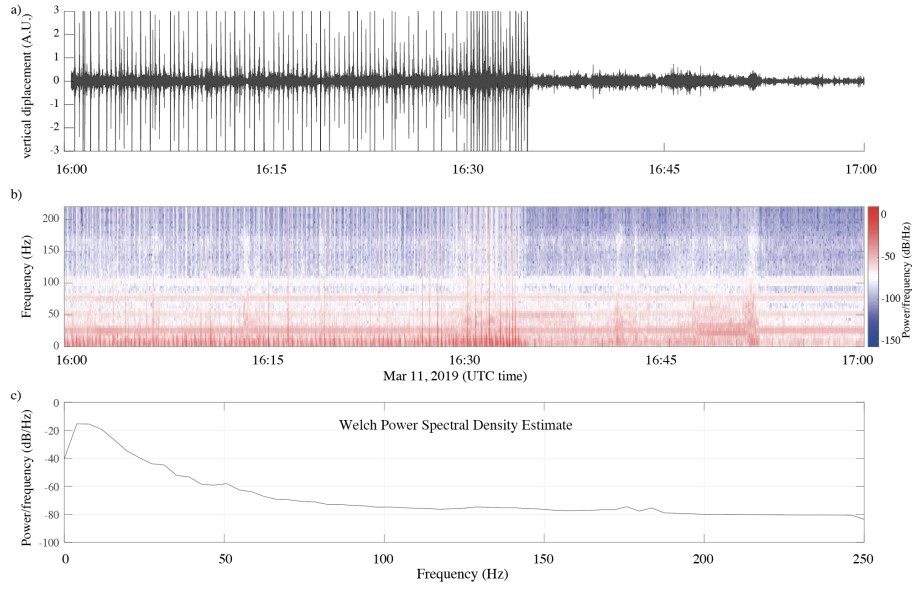

**Figure A2.** Same as figure S1, but in the afternoon, with human activity

## Appendix B: Covariance between the inverted parameters

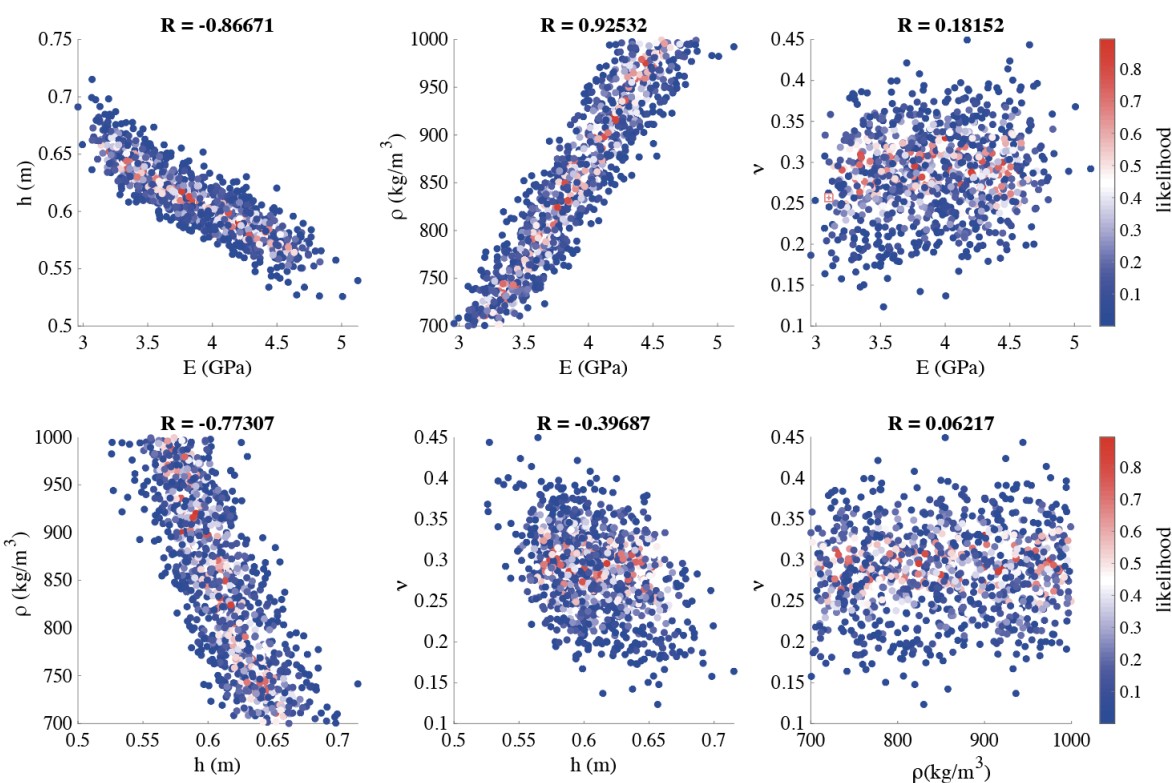

**Figure B1.** Covariance between Young's modulus, thickness, density and Poisson's ratio, with the associated correlation coefficient

*Author contributions.* Ludovic Moreau designed and led the Icewaveguide experiment in the scope of the IWG project (ANR10 LABX56). He supervised all this manuscript and research. Ludovic Moreau, Agathe Serripierri, Pierre Boue and Jérome Weiss participated in the deployment of the seismic array. Agathe Serripierri processed the data. Pierre Boue provided the code for computing correlations and beamforming. Philippe Roux contributed to the development of the methodology and supervised this research. Jérôme Weiss contributed to the interpretation of the results.

*Competing interests.* The authors declare that they have no conflict of interest.

*Acknowledgements.* ISTerre is part of Labex OSUG@2020 (ANR10 LABX56). This research was funded by the Agence Nationale de la Recherche (ANR, France) and by the Institut Polaire Français Paul-Emile Victor (IPEV). All data used for this research are from the network with FDNS code XG (Moreau and RESIF, 2019).

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
