# Peer review of "Recovering and monitoring the thickness, density and elastic properties of sea ice from seismic noise recorded in Svalbard"

_The Cryosphere, 2021_

## Community Comment (CC1)

**Answer to reviewer 1**

This paper presents a novel seismic approach aiming to temporally monitor the formation of sea ice through the deployment of a dense nodal seismic network. Thorough processing of ambient noise through beamformed cross-correlations resulted in the recovery of waveguided modes that were subsequently used as the basis for a Bayesian inverse scheme.

The paper is well written and constructed, as are most coming out of this group, and is suitable for publication upon very minor revisions. My few questions/comments are as follow.

We thank the reviewer for the constructive feedback. We have answered the comments below, and carefully reviewed our manuscript. In accounting for the comments of the reviewers, we have added a figure in section 2.2.2 to describe the workflow for recovering the noise correlation function. We also added two figures in Appendix A to show the characteristics of the recorded seismic noise, and a figure in Appendix B to show the covariance of sea ice parameters.

1) On paragraph 165: consider describing a bit better in math the procedure related to the SVD decomposition of the FK transform.

We have added a more detailed description of the processing. However, it is complicated to add more mathematical details without going into the full description of the method. Those details can be found in Minonzio et al. (2010).

This processing consists in the following steps:

1. The matrix of transmit-receive signals has three-dimensions: sources (M = 2 or 4), receivers (N), and time. The first step is the application of the Fourier transform to the temporal dimension of this matrix.

2. At each frequency, the resulting Fourier-domain matrix is sliced into 2D transmit-receive matrices. These matrices are then decomposed into singular values. The singular vectors define an orthonormal basis of the space dimensions along the transmitters (left-singular vector) and receivers (right-singular vector). The underlying idea behind this processing step is that the different levels of modal energy are distributed onto the singular vectors, the energy information being contained in the singular values. This allows a heuristic separation of the noise and signal subspaces, in a classical way for singular value-based filters.

3. The last step consists of defining test vectors that are representative of the wave propagation problem. In the present case, we use plane waves of the form $e^{-ik_{test}x_n}$, $k_{test}$ is the wavenumber to be tested, and $x_n$ (n = 1,2,...,N) is the coordinate of receiver n along the propagation. Finally, the test vectors are projected onto the singular vectors of the receivers' basis. This leads to a scalar product that is maximized when the wavenumber in the test vector matches that of the waves in the measured wavefield. In practice, this projection step is equivalent to calculating the discrete spatial Fourier transform of each singular vector.
Step 3 is performed at each frequency resulting from step 1.

Once steps 1-3 are performed, the resulting frequency-wavenumber spectrum significantly enhances the identification of the dispersion curves, for two reasons: (i) it is possible to separate signal from noise by applying a threshold to the singular values; and (ii) modal amplitude stands out at all frequencies and for all modes with the same spectral intensity (Fig. 4), despite their different relative amplitude in the wavefield, because singular vectors have a unit norm. The dispersion curves can therefore be identified on a larger bandwidth and with less SNR-related uncertainties than with conventional beamforming techniques (Moreau, Boué et al, 2020).

2) Could additional dispersion information have been retrieved by simply picking the maximum of the beamformer at every frequency?

This is a good question, because beamforming is known to be very robust to SNR. However, it is limited by spatial sampling. To apply beamforming, the full 2D array should be used. In the 2D array, spatial sampling is 4 meters, which allows to beamform the data without aliasing only up to 16 Hz (Moreau, Boué et al, 2020). Moreover, the SVD processing accounts for the multiplicity of sources with much more sensitivity (see figure B1 in Moreau, Boué et al, 2020). Hence, no additional information could be retrieved from beamforming, it is actually the opposite.

We have added the following sentence in section 2.2.2.

"Applying beamforming at frequencies higher than 16 Hz would cause aliasing problems, which prevents dispersion information to be extracted beyond 16 Hz."

We have also added a statement after the more detailed description of the SVD-based dispersion curves extraction:

"The dispersion curves can therefore be identified on a larger bandwidth and with less SNR-related uncertainties than with conventional beamforming techniques (Moreau, Boué et al, 2020)."

3) Why pose as an MCMC instead of a full grid search? In your case, your forward model is purely analytical unless I am mistaken, which means that a quick parallel implementation of a grid search should be quite feasible.

We understand from this question that the reviewer is asking about the efficiency of parallelized computation versus that of MCMC, which is sequential and thus cannot be parallelized. There are three main reasons why we have chosen to use MCMC:

1- When using a standard grid search, accuracy depends on the grid size. When using MCMC, the steps in the Markov Chain depend on the likelihood of the position in the parameters space. Typically, positions with high likelihood will be scanned with a much finer resolution than positions with low likelihood. For example, to achieve the same resolution with a grid search, one should use a grid size of 0.01 m for the thickness, 0.02 GPa for the Young's modulus, 0.02 for the Poisson's ratio and 40 kg for the density. This would require about 240 000 combinations to search through a parameters space of the same dimension as the one in the manuscript. With MCMC we require half this amount. Of course, a parallelized implementation of the grid search would still be much faster, but the carbon footprint of the inversion would be at least twice.
2- The main interest for Bayesian inference instead of a grid search is to obtain statistical information about the inverted parameters. The MCMC output is the joint probability density function of the parameters.
3- The simulated annealing part of our MCMC implementation allows the variance of the measurement errors to be estimated, which is a useful information when trying to identify the origin of potential discrepancies in the results. For example, in the present case the pattern shown in figure 5 tends to indicate that ice properties might vary over time more quickly in the NS direction than they do in the EW direction.

4) I'm curious as to why things seem to be generally insensitive to density, since this is an important stated objective of your study. I would recommend you explaining what you mean by "The inversions give a value that is very stable, at around $910\pm82$ kg.m$^{-3}$ for the EW direction, and $908 \pm 80$ kg.m$^{-3}$ for the NS direction" on 320, and a second (perhaps unintentionally repeated) time: "Nevertheless, the actual uncertainty is probably much smaller, as all of the inversions give a very stable value" on 325, and again . The posterior density is more or less flat, and thus the mean value here, regardless of its consistency across lines, should probably not be treated as a constrained parameter.

Perhaps an added statement as to why you think the inversion fails to robustly constrain density might be helpful, or in what ways it could be accounted for.

We agree that density is not well-constrained. We believe that the larger standard deviation reflects the limits of our forward model more than it is an indicator of the limits of the methodology. The fact that the estimated density remains constant through days and directions is an indicator that this assumption is very likely.

The model is only sensitive to the effective properties of the {ice+snow} system, because it cannot account for the snow layer (about 40 cm thick, on average), which modifies the effective properties. Intuitively, it appears that the weight of the snow layer modifies the density of the {ice+snow} system more than it does its rigidity (Young's modulus) and expansion/contraction (Poisson's ratio). Presumably, a forward model able to account for snow would be a significant improvement, which should constrain the density in a better way.
We have added the following statement at the end of section 2.4.2.

Interestingly, the covariance of the parameters (see figure B1 in Appendix B) indicates that Poisson's ratio, despite being well-constrained, seems rather uncorrelated from the other parameters. On the other hand, Young's modulus, density and thickness appear to be strongly correlated, despite the density being not very well-constrained.

The observations seem to indicate that density having a flatter PDF reflects the limits of our forward model more than it is an indicator of the limits of the methodology. The model is only sensitive to the effective properties of the {ice+snow} system, because it cannot account for the snow layer (about 40 cm thick, on average), which modifies the effective properties. The weight of the snow layer modifies the density of the {ice+snow} system more than it does its rigidity (Young's modulus) and expansion/contraction (Poisson's ratio). Presumably, a forward model able to account for snow would be a significant improvement, which should constrain the density in a better way. The development of such a forward model is therefore an important follow up of this work.

[Figure]

Figure B1. Covariance between Young's modulus, thickness, density and Poisson's ratio, with the associated correlation coefficient

---

## Community Comment (CC2)

**Answer to reviewer 2**

The authors present an innovative passive seismic study that monitors the evolution of sea ice thickness and mechanical properties using passive seismic noise. The authors combine both array processing with ambient noise cross-correlation to produce dispersion curves for guided waves within sea ice. The results of these dispersion measurements are used to invert for ice thickness, Young's modulus, Poisson's ratio and ice density using a Bayesian framework.

The manuscript is interesting and merits publication in The Cryosphere. It is concise and well written but I think improvements could be made to the clarity of the paper. I have a number of minor comments and suggestions which are detailed below.

We thank the reviewer for the constructive comments. We hope the changes made to the manuscript, together with our answers to the comments, will help clarify the paper. In accounting for the comments of the reviewers, we have added a figure in section 2.2.2 to describe the workflow for recovering the noise correlation function. We also added two figures in Appendix A to show the characteristics of the recorded seismic noise, and a figure in Appendix B to show the covariance of sea ice parameters.

- Section 2.1: Seismic array would benefit from a paragraph describing the deployment of the seismic instruments, with details on how the geophones were installed i.e were they mounted prior to deployment, burial depth if at all and any challenges or difficulties encountered.

We have added the following paragraph in section 2.1 to describe the practical aspects of the deployment

"The type of geophones used for the experiment has a cylindrical geometry of about 17 cm in height and 12 cm in diameter, mounted on a detachable spike. The geophones were installed directly in the ice without their spike. For accurate positioning, we used taut cords that we attached to each end of the rows and columns of the array, and a decameter. To maximize the coupling, a milling tool was specifically designed to drill the ice at the diameter of the nodes. The snow was removed prior to drilling holes, and geophones were installed in the holes at about half their height. We covered them back with snow to insulate them in view of preserving their battery life. At the time of the deployment, the internal temperature of several nodes was measured, before and after covering them with snow, showing an increase from -21 to -16ºC. Deployment took about 2.5 days of work for a team of five people, including the time required for their activation. Markers were carefully placed all around the main array and at the position of the four antennae, in order to find them back more easily at the end of the experiment."

- Section 2.2.1 would benefit from a few sentences on the theoretical foundations on the retrieval of Green's functions from ambient noise. This section should also include a description of the processing steps, parameters used and the justification for their selection e.g., why and how was a 5 minute window selected for the noise correlation length. Also, which stations were used and if correlations were made across stations components e.g., N with E.

Theoretical foundations of on the retrieval of Green's functions from ambient noise can be found in [P. Roux, K. G. Sabra, W.A. Kuperman and A. Roux (2005)"Ambient noise cross-correlation in free space: theoretical approach", J. Acoust. Soc. Am. 117(1), pp. 79-84] and [Campillo and Roux, "Seismic imaging and monitoring with ambient noise correlations", Treatise on Geophysics, second Edition, Vol. 1, Edited by B. Romanowicz and A. Dziewonski, Elsevier-Amsterdam, 256-271, 2014], which have been added to the reference list.

As written in the paper (lines 120-125), a homogenized wave field illuminating the propagation medium in all directions is a prerequisite for obtaining an accurate Green's function estimation. For seismic data recorded on glaciers, this condition imposes strong limitations on Green's function convergence because of minimal seismic scattering in homogeneous ice and limitations in network coverage.

We have added the following text to section 2.2.1

Recent work provides a catalogue of methods to tackle the challenge of applying passive seismic interferometry to glaciers in the absence of isotropic source distribution Sergeant et al. (2020).

The remark concerning the 5-minutes window cannot be treated lightly and requires additional figures. We have added an Appendix to the manuscript to include these figures (see figures A1 and A2 below). We have also added the following text to section 2.2.1

"Figures A1 and A2 show the seismic wavefield recorded on 11 March 2019, during the night (figure A1-a) and during the day (figure A2-a) when some fieldwork occurred about 500 m N-E of the main array. The corresponding short-time Fourier transforms are shown in figures S1-b and S2-b, as well as the estimated power spectral densities (figures A1-c and A2-c). These figures show that seismicity can be dominated by noise sources with very different characteristic time. For example, in presence of human activity the characteristic time is a few minutes (see figure A1-a between 16:00 and 16:35). When there is no human activity, noise sources can be impulsive when icequakes occur (there are also periods of time where many icequakes occur every minute), or they can have a characteristic time of a few minutes (see for example figure S1-a between 2:00 and 2:15).

Therefore, it is necessary to correlate noise segments with a length that accounts for the characteristic time of the various noise sources. After some preliminary tests, we concluded that a 5-minutes window is adequate. Moreover, impulsive events, human activity and other noise sources have different levels of energy. To prevent bias due to the dominant noise sources, spectral whitening was applied to the noise segments prior to calculate the correlations. Three sets of correlations were calculated:

- correlations between the recordings from the vertical displacement component

- correlations between the recordings from the north displacement component

-correlations between the recordings from the east displacement component

In this work, we restrict the study to correlations between (i) the 45 stations of the EW line with the 4 stations of the LEA and LAW, and (ii) the 45 stations of the NS line with the 4 stations of the LAS and LAN. Of course, by using all stations pairs amongst the 247 stations, it will be possible to obtain 30381 inter-correlations with many different inter-station paths, distances, and directions. With such amount of correlations, it becomes possible to apply tomographic inversions over the full array geometry, and to obtain a 3D+time map of the sea ice properties with unprecedented spatial and temporal resolution. However, this requires a different processing strategy where noise sources have to be selected so that their distribution around the array is isotropic. This is an on-going work that will be the matter of a future paper."

[Figure]

Figure A1 - a) One hour of seismic wavefield at night, on 11 March 2019, with b) the corresponding short time Fourier transform, and c) Welch Power Spectral density of the recording.

[Figure]

Figure A2 – Same as figure A1, but in the afternoon, with human activity.

- Section 2.2.2: In this section, it is unclear as to which of the seismic instruments (1C or 3C) is used in the beamforming and subsequent calculation of the noise correlation function. I assume that the authors beamform seismic noise using the 1C stations before using noise directed within 10 degrees of the 2 3C lines to compute the correction function using the 3C stations.

Actually, beamforming was performed with all stations that are equally spaced within the main array, which also includes 3C stations. To clarify that, we have added the following sentence in this section.

"Beamforming was performed using all 1C and 3C stations of the main array that are equally spaced".

We have also added, in section 2.2.1 (see the answer of your previous question), more details about which stations are being used for the correlations.

- Section 2.2: A flow chart summarizing the beamforming and noise correlation function workflow should be included to improve the clarity of this section.

We have added a flow chart as requested. It is now in figure 4.

- Section 2.3: The authors should expand upon the SVD methodology used to compute the dispersion curves.

Reviewer 1 also requested more details about the SVD-based dispersion curves extraction. We have added more details to describe the method, without going too much into the mathematical details, which are given in Minonzio et al. (2010). The following description was included.

This processing consists in the following steps:

1. The matrix of transmit-receive signals has three-dimensions: sources (M = 2 or 4), receivers (N), and time. The first step is the application of the Fourier transform to the temporal dimension of this matrix.

2. At each frequency, the resulting Fourier-domain matrix is sliced into 2D transmit-receive matrices. These matrices are then decomposed into singular values. The singular vectors define an orthonormal basis of the space dimensions along the transmitters (left-singular vector) and receivers (right-singular vector). The underlying idea behind this processing step is that the different levels of modal energy are distributed onto the singular vectors, the energy information being contained in the singular values. This allows a heuristic separation of the noise and signal subspaces, in a classical way for singular value-based filters.

3. The last step consists of defining test vectors that are representative of the wave propagation problem. In the present case, we use plane waves of the form $e^{-ik_{test}x_n}$, where $k_{test}$ is the wavenumber to be tested, and $x_n$ (n = 1,2,...,N) is the coordinate of receiver n along the propagation. Finally, the test vectors are projected onto the singular vectors of the receivers' basis. This leads to a scalar product that is maximized when the wavenumber in the test vector matches that of the waves in the measured wavefield. In practice, this projection step is equivalent to calculating the discrete spatial Fourier transform of each singular vector.
Step 3 is performed at each frequency resulting from step 1.

Once steps 1-3 are performed, the resulting frequency-wavenumber spectrum significantly enhances the identification of the dispersion curves, for two reasons: (i) it is possible to separate signal from noise by applying a threshold to the singular values; and (ii) modal amplitude stands out at all frequencies and for all modes with the same spectral intensity (Fig. 4), despite their different relative amplitude in the wavefield, because singular vectors have a unit norm. The dispersion curves can therefore be identified on a larger bandwidth and with less SNR-related uncertainties than with conventional beamforming techniques (Moreau, Boué et al, 2020).

[Figure]

Figure 4. Workflow to calculate the noise correlation function from seismic noise.

- Section 2.4.2: In the inversion, why haven't the author's used uncertainties from their measurements for the estimate of sigma in equation 6? It is unclear from the text why this isn't possible. I'm also unsure as to why the Simulated Annealing method is used instead of the burn-in phase of the MCMC. Could this not affect the sampling of the posterior distribution?

We agree with the reviewer that this may look unclear. The problem when trying to estimate sigma comes from the fact that it should account for uncertainties in the dispersion curves, not those in the measurements. There is no linear relationship between both, because of the dispersion of the QS mode. That is what we had tried to explain in section 2.4.2 when writing:

"The variance is linked to measurement errors; i.e., a mix between the influence of the SNR and other random perturbations of the measure, such as variations in the physics of the problem. In the present problem, such perturbations are averaged when going from the time-space to the frequency-wavenumber domains, and thus the variance reflects variations of frequency- wavenumber values around ground truth values."

Moreover, depending on the quality of the correlation function, these variations are not the same between all sets of dispersion curves. This is why we have chosen to estimate sigma with the simulated annealing algorithm. In an attempt to make this clearer in the manuscript, we have added the following explanations.

"However, $\sigma^2$ is data-dependent. In our approach, we are inverting dispersion curves, so it should account for uncertainties in the dispersion curves, not those in the measurements. Unfortunately, there is no linear relationship between both, because of the dispersion of the QS mode. Depending on the quality of the correlation function, these uncertainties are not the same between all sets of dispersion curves. Hence $\sigma^2$ cannot be set *a priori* for all of the inversions in this study, or else it would be a coarse approximation."

The posterior distribution is actually approximated after the burn-in phase, when the MCMC algorithm has sufficiently explored the solution space. The quality of the estimated posterior distribution depends essentially on the number of samples generated. It does not depend on the position where the algorithm is initialized (which also determines the burn-in phase). Figure 6 shows that the posterior distributions are well-estimated over the whole range of values in the search space.

- Results and Discussion: This section would benefit from comparisons between the retrieved parameters and previous estimates in past studies e.g., How does the Young's modulus compare with other estimates? Likewise, how does the estimated value of density compare with meteoric ice and sea ice?

We have added the following discussion regarding values found in the literature for sea ice properties. This discussion appears after Eq. (6) to justify the choice of the parameters space.

For first year ice, in situ measurements of density range from 840 to 910 kg/m3 for the ice above the waterline, and 900 to 940 kg/m3 for the ice below the waterline (Timco and Frederking, 1996). Poisson's ratio varies between 0.25 and 0.4, Young's modulus between 2 and 6 GPa (Anderson, 1958; Timco and Weeks, 2010). In particular, in the Van Mijen fjord, Romeyn et al. (2021) reported a Young's modulus around 2.5 GPa with a Poisson's ratio of 0.33, and Morozov et al (2012) reported a Young's modulus around 3 GPa and a Poisson's ratio of 0.3. (Moreau et al, 2014c), reported a Young's modulus around 4 GPa and a Poisson's ratio of 0.32 at Vallunden. The slightly higher value of Young's modulus at Vallunden, in comparison with those directly in the Van Mijen fjord, is likely attributable to the protected physical setting of the study site, and support from the surrounding shoreline at the moraine. Regarding the thickness, ice drillings indicated that it was systematically less than 1 m. Hence, we assume for the prior distributions that the model parameters have equal probability over a range of values that contains the above-referenced values.

We also refer to this discussion in the "Results and discussion" section.

The estimations are consistent with the values reported in the literature for first year ice, which are given in section 2.4.2.

Additionally, the inversion appears to be insensitive to density. Why do you think this is the case? How could the inversion be improved to account for the insensitivity to density?

Reviewer 1 also requested a discussion about the density. We have added the following discussion.

Interestingly, the covariance of the parameters (see figure B1 in Appendix B) indicates that Poisson's ratio, despite being well-constrained, seems rather uncorrelated from the other parameters. On the other hand, Young's modulus, density and thickness appear to be strongly correlated, despite the density being not very well-constrained.

The observations seem to indicate that density having a flatter PDF reflects the limits of our forward model more than it is an indicator of the limits of the methodology. The model is only sensitive to the effective properties of the {ice+snow} system, because it cannot account for the snow layer (about 40 cm thick, on average), which modifies the effective properties. The weight of the snow layer modifies the density of the {ice+snow} system more than it does its rigidity (Young's modulus) and expansion/contraction (Poisson's ratio). Presumably, a forward model able to account for snow would be a significant improvement, which should constrain the density in a better way. The development of such a forward model is therefore an important follow up of this work.

[Figure]

Figure B1. Covariance between Young's modulus, thickness, density and Poisson's ratio, with the associated correlation coefficient

- Figure 1: A map of Svalbard and the location of Lake Vallunden highlighted should be included.

A small map has been added in Figure 1a to show the position of the experiment in Svalbard

- Figure 6: The histograms should be normalized such that the area under each graph is 1 in order to show the PDF for each of the parameters. It is difficult to see the blue line indicating the maximum of each distribution and this should be changed to black or the shading of the histograms should be changed to grey or white.

These modifications have been made in Figure 6.

- Lines 78 - 81: A schematic diagram illustrating the displacements of the different modes of the waveguides would be beneficial.

We are showing this figure here, in response to the reviewer. However, we believe that it is not necessary for the manuscript to be self-consistent. It would be difficult to add such a figure in the manuscript without significantly modifying its structure, because this would require to elaborate on the theory of guided waves. This was already made in Moreau, Boué et al (2020), and we would rather build up from this work than to repeat it.

[Figure]

a) Displacement field generated by the *QS* mode (top) and the *QS₀* mode (bottom) in a floating ice layer of thickness $h$=50 cm at a frequency of 100Hz, with $E$= 4.5 GPa, $\nu$= 0.33, and $\rho$= 910km/m$^3$. b) Wavenumber-thickness versus frequency-thickness dispersion curves of the ice layer. Black dots show the theoretical values and red lines the approximation of the *QS* and *QS₀* modes by the flexural and axial waves (Stein et al, 1998), which is valid when the frequency-thickness product is less than 50 Hz·m.

- Lines 280 - 281: Please include more detail on how the distribution is fitted to the posterior distribution.

We have added the following description

After the MCMC algorithm has completed, a PDF is generated for E, h, ν, and ρ. Figure 6 shows an example of PDF obtained on March 10, from line EW. The ice properties are determined by computing the histograms of the parameters, to which we fit with a kernel-type distribution density estimator. The kernel estimation method creates a smooth, continuous curve to represent the probability distribution of the data from the data samples by estimating locally the normal distribution function centered in each sample (the kernel functions). Summing these local smoothing functions for each sample produces the resulting continuous fit. We use the maximum of this fit to determine the estimate of the parameters.

- Lines 284 - 285: Since the distributions are not Gaussian, I'm unsure how relevant it is quoting uncertainties for the results using the standard deviation. The interquartile range would be a more relevant measure of uncertainty.

One may argue that it would also be possible to fit a gaussian to the posterior distributions. We have chosen a kernel-type fit only for more accurate values around the maxima. For this reason, we would rather keep a standard deviation as a measure of the dispersion of the PDF. This also allows us to remain consistent with the definition of the likelihood function.

---

## Community Comment (CC3)

**Answer to reviewer 3**

This is a very interesting analysis of a uniquely dense sea ice deployment of seismographs using ambient noise guided mode dispersion curves to invert for bulk ice properties (in two propagation directions). With a bit more detail (suggestions below) it's suitable for publication in its present form as an initial analysis of a data set that I strongly suspect has a lot more to offer in more detailed analysis. Will the data be publicly available somewhere (not apparently in the Acknowledgments).

Most of my comments below address where I suggest that methodological and other points might be usefully expanded (including, perhaps, some supplemental figures that could be in a appendix).

We thank reviewer for the positive feedback and all the suggestions and comments to improve the manuscript. Our answers are given thereafter. In accounting for the comments of the reviewers, we have added a figure in section 2.2.2 to describe the workflow for recovering the noise correlation function. We also added two figures in Appendix A to show the characteristics of the recorded seismic noise, and a figure in Appendix B to show the covariance of sea ice parameters.

The authors have done a good job in fitting analytical expressions to (beamform-selected) autocorrelations, with modes decomposed via a SVD method, to extract dispersion curves in EW and NS directions. The size and density of the array used (247 seismographs with interstation spacings as close as 1 m) is unprecedented to my knowledge for this type of application. Additional deployment and experimental details would be appreciated. Were the instruments buried, for example? Were ice cores taken for ice-truth measurements during of the experiment? How long did it take to deploy these instruments at this scale (important to know if such experiments might be repeated in other locations).

Reviewer 2 also requested more practical information about the deployments. Most of these details are available in Moreau, Boué, et al. (2019). However, we have added the following text to make the manuscript more self-consistent.

"The type of geophones used for the experiment has a cylindrical geometry of about 17 cm in height and 12 cm in diameter, mounted on a detachable spike. The geophones were installed directly in the ice without their spike. For accurate positioning, we used taut cords that we attached to each end of the rows and columns of the array, and a decameter. To maximize the coupling, a milling tool was specifically designed to drill the ice at the diameter of the nodes. The snow was removed prior to drilling holes, and geophones were installed in the holes at about half their height. We covered them back with snow to insulate them in view of preserving their battery life. At the time of the deployment, the internal temperature of several nodes was measured, before and after covering them with snow, showing an increase from -21 to -16$^{\circ}$C. Deployment took about 2.5 days of work for a team of five people, including the time required for their activation. Markers were carefully placed all around the main array and at the position of the four antennae, in order to find them back more easily at the end of the experiment."

Could similar (or at least practically useful) results be obtained with a smaller deployment (this can be tested by decimating the data set of NCFs). This would be valuable to address given the motivations expressed in the abstract and paper.

This is an important point. We have demonstrated in Moreau, Weiss et al. (2019) that results with the same accuracy can be obtained by exploiting the waveforms of the icequakes recorded with as few as 3-5 stations. The methodology differs significantly from that in this manuscript, since it does not require to calculate the dispersion curves of the guided modes. A systematic inversion of the thousands of icequakes recorded during deployment will provide much more information about the spatial variations of sea ice parameters, that will not be limited to lines NS and EW. However, this requires to extract all the icequakes from the recordings.

This is an on-going work where deep learning strategies are being implemented to this end. Here we share some exciting upcoming results (see figure below), where part of these icequakes have been extracted and inverted for the source position as well as the ice thickness, using 5 stations only. Icequakes were recorded for the whole duration of the deployment.

[Figure]

Icequakes (colored dots) extracted and inverted for source position and average ice thickness between source and the 5 stations used for inversion (shown as black squares). Icequakes were recorded between 27 February and 24 March 2019. Note the change in thickness for icequakes coming from the same position, which indicates that ice has grown during deployment. The inverted ice thicknesses are consistent with those found using the EW and NS array lines and the processing introduced in the present paper. The location of the icequakes follow the shore line, where cracks have been visually identified. These cracks were likely cause by thermal and tidal forcing.

Complicating methodological or physical factors that may explain the observed variance in the parameters were underexplored (e.g., can the dispersion curves be better resolved or meaningfully smoothed to improve the data as shown in example in Figure 4?).

Reviewer 1 also asked if it would be possible to improve the retrieval of the dispersion curves. We have elaborated on the SVD-based processing to explain why it is unlikely to obtain better results from currently available alternative strategies. This is essentially due to the fact that the SVD already accounts for the multiplicity of virtual sources (i.e. the four stations in each of the four linear arrays to the east, west, south and north) in an optimal way. We have added the following explanations.

This processing consists in the following steps:

1. The matrix of transmit-receive signals has three-dimensions: sources (M = 2 or 4), receivers (N), and time. The first step is the application of the Fourier transform to the temporal dimension of this matrix.

2. At each frequency, the resulting Fourier-domain matrix is sliced into 2D transmit-receive matrices. These matrices are then decomposed into singular values. The singular vectors define an orthonormal basis of the space dimensions along the transmitters (left-singular vector) and receivers (right-singular vector). The underlying idea behind this processing step is that the different levels of modal energy are distributed onto the singular vectors, the energy information being contained in the singular values. This allows a heuristic separation of the noise and signal subspaces, in a classical way for singular value-based filters.

3. The last step consists of defining test vectors that are representative of the wave propagation problem. In the present case, we use plane waves of the form $e^{-ik_{test}x_n}$, where $k_{test}$ is the

wavenumber to be tested, and $x_n$ (n = 1,2,...,N) is the coordinate of receiver n along the propagation. Finally, the test vectors are projected onto the singular vectors of the receivers' basis. This leads to a scalar product that is maximized when the wavenumber in the test vector matches that of the waves in the measured wavefield. In practice, this projection step is equivalent to calculating the discrete spatial Fourier transform of each singular vector.
Step 3 is performed at each frequency resulting from step 1.

Once steps 1-3 are performed, the resulting frequency-wavenumber spectrum significantly enhances the identification of the dispersion curves, for two reasons: (i) it is possible to separate signal from noise by applying a threshold to the singular values; and (ii) modal amplitude stands out at all frequencies and for all modes with the same spectral intensity (Fig. 4), despite their different relative amplitude in the wavefield, because singular vectors have a unit norm. The dispersion curves can therefore be identified on a larger bandwidth and with less SNR-related uncertainties than with conventional beamforming techniques (Moreau, Boué et al, 2020).

It was unclear (to me) exactly which stations were being correlated for use in constraining the NCFs (e.g., the linear array stations and azimuthally selected stations in the grid (which would mean that only stations intersecting a 10-degree cone from the linear array stations were actually utilized, but no stations from the corners, for example (?)). It would seem that a great many potential cross-correlations within the experiment were not used. That's fine for this initial study, but some additional detail and quantification of this geometry and data usage would be appreciated (with 247 stations there are potentially around 30,000 potential NCFs that could be calculated, of course; how many total NCFs went into the inversions here out of these potential ~30,000 station pairs?). I'd suspect that a more comprehensive subarray strategy might yield improved results, and the ability to assess spatial variations (the authors hint that this will be a next step, along with icequake analysis in future work at the end of the paper). Structural variation across the study region may explain some of this four-parameter estimation experiment (but again it's not clear to me but it would be nice to assess this effect more thoroughly; it's alluded to for example in line 270).

The reviewer is right, and much more information about the ice could be obtained by using all station-pair combinations to calculate the correlations. We have added the following explanation to justify why we restrict the study to the stations of the dense cross and the linear arrays only.

In this work, we restrict the study to correlations between (i) the 45 stations of the EW line with the 4 stations of the LEA and LAW, and (ii) the 45 stations of the NS line with the 4 stations of the LAS and LAN. Of course, by using all stations pairs amongst the 248 stations, it will be possible to obtain 30381 inter-correlations with many different inter-station paths, distances, and directions. With such amount of correlations, it becomes possible to apply tomographic inversions over the full array geometry, and to obtain a 3D+time map of the sea ice properties with unprecedented spatial and temporal resolution. However, this requires a different processing strategy where noise sources have to be selected so that their distribution around the array is isotropic. This is an on-going work that will be the matter of a future paper.

The authors employ a simulated annealing strategy to obtain a starting model. It was unclear to me, at least, how this leads to new information regarding determination of the measurement errors, unless this was simply evaluated to be consistent with the chi-square value, which is simply consistent). For this reason. I suggest that it is more accurate to indicate that the fit (e.g., line 280) between model and data is statistically "consistent", rather than "excellent".

In a way, we are indeed attempting to be consistent with the chi-square value, so we have replaced "excellent fit" with "statistically consistent fit". Reviewer 2 also asked about the interest of the simulated annealing phase. Here we give the same answer.

We agree with the reviewer that this may look unclear. The problem when trying to estimate sigma comes from the fact that it should account for uncertainties in the dispersion curves, not those in the measurements. There is no linear relationship between both, because of the dispersion of the QS mode. That is what we had tried to explain in section 2.4.2 when writing:
"The variance is linked to measurement errors; i.e., a mix between the influence of the SNR and other random perturbations of the measure, such as variations in the physics of the problem. In the present problem, such perturbations are averaged when going from the time-space to the frequency-wavenumber

domains, and thus the variance reflects variations of frequency- wavenumber values around ground truth values."

Moreover, depending on the quality of the correlation function, these variations are not the same between all sets of dispersion curves. This is why we have chosen to estimate sigma from the simulated annealing part.

In an attempt to make this clearer in the manuscript, we have added the following explanations.

"However, $\sigma^2$ is data-dependent. In our approach, we are inverting dispersion curves, so it should account for uncertainties in the dispersion curves, not those in the measurements. Unfortunately, there is no linear relationship between both, because of the dispersion of the QS mode. Depending on the quality of the correlation function, these uncertainties are not the same between all sets of dispersion curves. Hence $\sigma^2$ cannot be set *a priori* for all of the inversions in this study, or else it would be a coarse approximation."

They subsequently use MCMC to obtain a posterior PDF with a uniform (limited) range prior distribution for each parameter, but do not show the covariances of the parameters. A posterior illustration of parameter covariances would also be helpful to illustrate the tradeoff space between the four parameters.

The range of prior distribution was chosen so as to include values found in literature for first year ice. We have added the following justification:

For first year ice, in situ measurements of density range from 840 to 910 kg/m3 for the ice above the waterline, and 900 to 940 kg/m3 for the ice below the waterline (Timco and Frederking, 1996). Poisson's ratio varies between 0.25 and 0.4, Young's modulus between 2 and 6 GPa (Anderson, 1958; Timco and Weeks, 2010). In particular, in the Van Mijen fjord, Romeyn et al. (2021) reported a Young's modulus around 2.5 GPa with a Poisson's ratio of 0.33, and Morozov et al (2012) reported a Young's modulus around 3 GPa and a Poisson's ratio of 0.3. (Moreau et al, 2014c), reported a Young's modulus around 4 GPa and a Poisson's ratio of 0.32 at Vallunden. The slightly higher value of Young's modulus at Vallunden, in comparison with those directly in the Van Mijen fjord, is likely attributable to the protected physical setting of the study site, and support from the surrounding shoreline at the moraine. Regarding the thickness, ice drillings indicated that it was systematically less than 1 m. Hence, we assume for the prior distributions that the model parameters have equal probability over a range of values that contains the above-referenced values.

We have added a new figure in Appendix B (see below), which shows the covariance between parameters. We have also modified the comments regarding the PDF accordingly:

Interestingly, the covariance of the parameters (see figure B1 in Appendix B) indicates that Poisson's ratio, despite being well-constrained, seems rather uncorrelated from the other parameters. On the other hand, Young's modulus, density and thickness appear to be strongly correlated, despite the density being not very well-constrained.

The observations seem to indicate that density having a flatter PDF reflects the limits of our forward model more than it is an indicator of the limits of the methodology. The model is only sensitive to the effective properties of the {ice+snow} system, because it cannot account for the snow layer (about 40 cm thick, on average), which modifies the effective properties. The weight of the snow layer modifies the density of the {ice+snow} system more than it does its rigidity (Young's modulus) and expansion/contraction (Poisson's ratio). Presumably, a forward model able to account for snow would be a significant improvement, which should constrain the density in a better way. The development of such a forward model is therefore an important follow up of this work.

[Figure]

Figure B1. Covariance between Young's modulus, thickness, density and Poisson's ratio, with the associated correlation coefficient

It is interesting that the inferred ice thicknesses differ systematically in the NS and EW directions in Figure 7. Can the authors elaborate on this (in line 270 they allude to special variations in the physics of the problem, but I believe what they meant to indicate was something like "spatial variations in ice model parameters".

We meant "spatial variations" indeed. This has been modified. We do not have a clear explanation for this. A likely explanation could be related to the elongated geometry of the moraine in the NS direction, which could have a mechanical effect to the ice while it is growing. Also, the channel to the north, where water current is present could have an effect. However, it is noteworthy that the differences between both directions are not statistically significant, since the uncertainties overlap, so it could also be linked to the quality of the Green's function retrieval. This remark was added to the manuscript.

All things considered, this is a significant work with a unique data set that could use a bit more polishing to realize its potential, and will no doubt be valuable for further analysis.

---

## Author Comment (AC3)

**Answer to reviewer 3**
This is a very interesting analysis of a uniquely dense sea ice deployment of seismographs using ambient noise guided mode dispersion curves to invert for bulk ice properties (in two propagation directions). With a bit more detail (suggestions below) it's suitable for publication in its present form as an initial analysis of a data set that I strongly suspect has a lot more to offer in more detailed analysis. Will the data be publicly available somewhere (not apparently in the Acknowledgments).

Most of my comments below address where I suggest that methodological and other points might be usefully expanded (including, perhaps, some supplemental figures that could be in a appendix).

We thank reviewer for the positive feedback and all the suggestions and comments to improve the manuscript. Our answers are given thereafter. In accounting for the comments of the reviewers, we have added a figure in section 2.2.2 to describe the workflow for recovering the noise correlation function. We also added two figures in Appendix A to show the characteristics of the recorded seismic noise, and a figure in Appendix B to show the covariance of sea ice parameters.

The authors have done a good job in fitting analytical expressions to (beamform-selected) autocorrelations, with modes decomposed via a SVD method, to extract dispersion curves in EW and NS directions. The size and density of the array used (247 seismographs with interstation spacings as close as 1 m) is unprecedented to my knowledge for this type of application. Additional deployment and experimental details would be appreciated. Were the instruments buried, for example? Were ice cores taken for ice-truth measurements during of the experiment? How long did it take to deploy these instruments at this scale (important to know if such experiments might be repeated in other locations).

Reviewer 2 also requested more practical information about the deployments. Most of these details are available in Moreau, Boué, et al. (2019). However, we have added the following text to make the manuscript more self-consistent.

"The type of geophones used for the experiment has a cylindrical geometry of about 17 cm in height and 12 cm in diameter, mounted on a detachable spike. The geophones were installed directly in the ice without their spike. For accurate positioning, we used taut cords that we attached to each end of the rows and columns of the array, and a decameter. To maximize the coupling, a milling tool was specifically designed to drill the ice at the diameter of the nodes. The snow was removed prior to drilling holes, and geophones were installed in the holes at about half their height. We covered them back with snow to insulate them in view of preserving their battery life. At the time of the deployment, the internal temperature of several nodes was measured, before and after covering them with snow, showing an increase from -21 to -16°C. Deployment took about 2.5 days of work for a team of five people, including the time required for their activation. Markers were carefully placed all around the main array and at the position of the four antennae, in order to find them back more easily at the end of the experiment."

Could similar (or at least practically useful) results be obtained with a smaller deployment (this can be tested by decimating the data set of NCFs). This would be valuable to address given the motivations expressed in the abstract and paper.

This is an important point. We have demonstrated in Moreau, Weiss et al. (2019) that results with the same accuracy can be obtained by exploiting the waveforms of the icequakes recorded with as few as 3-5 stations. The methodology differs significantly from that in this manuscript, since it does not require to calculate the dispersion curves of the guided modes. A systematic inversion of the thousands of icequakes recorded during deployment will provide much more information about the spatial variations of sea ice parameters, that will not be limited to lines NS and EW. However, this requires to extract all the icequakes from the recordings.

This is an on-going work where deep learning strategies are being implemented to this end. Here we share some exciting upcoming results (see figure below), where part of these icequakes have been extracted and inverted for the source position as well as the ice thickness, using 5 stations only. Icequakes were recorded for the whole duration of the deployment.

[Figure]

Icequakes (colored dots) extracted and inverted for source position and average ice thickness between source and the 5 stations used for inversion (shown as black squares). Icequakes were recorded between 27 February and 24 March 2019. Note the change in thickness for icequakes coming from the same position, which indicates that ice has grown during deployment. The inverted ice thicknesses are consistent with those found using the EW and NS array lines and the processing introduced in the present paper. The location of the icequakes follow the shore line, where cracks have been visually identified. These cracks were likely cause by thermal and tidal forcing.

Complicating methodological or physical factors that may explain the observed variance in the parameters were underexplored (e.g., can the dispersion curves be better resolved or meaningfully smoothed to improve the data as shown in example in Figure 4?).

Reviewer 1 also asked if it would be possible to improve the retrieval of the dispersion curves. We have elaborated on the SVD-based processing to explain why it is unlikely to obtain better results from currently available alternative strategies. This is essentially due to the fact that the SVD already accounts for the multiplicity of virtual sources (i.e. the four stations in each of the four linear arrays to the east, west, south and north) in an optimal way. We have added the following explanations.

This processing consists in the following steps:

1. The matrix of transmit-receive signals has three-dimensions: sources (M = 2 or 4), receivers (N), and time. The first step is the application of the Fourier transform to the temporal dimension of this matrix.

2. At each frequency, the resulting Fourier-domain matrix is sliced into 2D transmit-receive matrices. These matrices are then decomposed into singular values. The singular vectors define an orthonormal basis of the space dimensions along the transmitters (left-singular vector) and receivers (right-singular vector). The underlying idea behind this processing step is that the different levels of modal energy are distributed onto the singular vectors, the energy information being contained in the singular values. This allows a heuristic separation of the noise and signal subspaces, in a classical way for singular value-based filters.

3. The last step consists of defining test vectors that are representative of the wave propagation problem. In the present case, we use plane waves of the form $e^{-ik_{test}x_n}$, where $k_{test}$ is the

wavenumber to be tested, and $x_n$ (n = 1,2,...,N) is the coordinate of receiver n along the propagation. Finally, the test vectors are projected onto the singular vectors of the receivers' basis. This leads to a scalar product that is maximized when the wavenumber in the test vector matches that of the waves in the measured wavefield. In practice, this projection step is equivalent to calculating the discrete spatial Fourier transform of each singular vector.
Step 3 is performed at each frequency resulting from step 1.

Once steps 1-3 are performed, the resulting frequency-wavenumber spectrum significantly enhances the identification of the dispersion curves, for two reasons: (i) it is possible to separate signal from noise by applying a threshold to the singular values; and (ii) modal amplitude stands out at all frequencies and for all modes with the same spectral intensity (Fig. 4), despite their different relative amplitude in the wavefield, because singular vectors have a unit norm. The dispersion curves can therefore be identified on a larger bandwidth and with less SNR-related uncertainties than with conventional beamforming techniques (Moreau, Boué et al, 2020).

It was unclear (to me) exactly which stations were being correlated for use in constraining the NCFs (e.g., the linear array stations and azimuthally selected stations in the grid (which would mean that only stations intersecting a 10-degree cone from the linear array stations were actually utilized, but no stations from the corners, for example (?)). It would seem that a great many potential cross-correlations within the experiment were not used. That's fine for this initial study, but some additional detail and quantification of this geometry and data usage would be appreciated (with 247 stations there are potentially around 30,000 potential NCFs that could be calculated, of course; how many total NCFs went into the inversions here out of these potential ~30,000 station pairs?). I'd suspect that a more comprehensive subarray strategy might yield improved results, and the ability to assess spatial variations (the authors hint that this will be a next step, along with icequake analysis in future work at the end of the paper). Structural variation across the study region may explain some of this four-parameter estimation experiment (but again it's not clear to me but it would be nice to assess this effect more thoroughly; it's alluded to for example in line 270).

The reviewer is right, and much more information about the ice could be obtained by using all station-pair combinations to calculate the correlations. We have added the following explanation to justify why we restrict the study to the stations of the dense cross and the linear arrays only.

In this work, we restrict the study to correlations between (i) the 45 stations of the EW line with the 4 stations of the LEA and LAW, and (ii) the 45 stations of the NS line with the 4 stations of the LAS and LAN. Of course, by using all stations pairs amongst the 248 stations, it will be possible to obtain 30381 inter-correlations with many different inter-station paths, distances, and directions. With such amount of correlations, it becomes possible to apply tomographic inversions over the full array geometry, and to obtain a 3D+time map of the sea ice properties with unprecedented spatial and temporal resolution. However, this requires a different processing strategy where noise sources have to be selected so that their distribution around the array is isotropic. This is an on-going work that will be the matter of a future paper.

The authors employ a simulated annealing strategy to obtain a starting model. It was unclear to me, at least, how this leads to new information regarding determination of the measurement errors, unless this was simply evaluated to be consistent with the chi-square value, which is simply consistent). For this reason. I suggest that it is more accurate to indicate that the fit (e.g., line 280) between model and data is statistically "consistent", rather than "excellent".

In a way, we are indeed attempting to be consistent with the chi-square value, so we have replaced "excellent fit" with "statistically consistent fit". Reviewer 2 also asked about the interest of the simulated annealing phase. Here we give the same answer.

We agree with the reviewer that this may look unclear. The problem when trying to estimate sigma comes from the fact that it should account for uncertainties in the dispersion curves, not those in the measurements. There is no linear relationship between both, because of the dispersion of the QS mode. That is what we had tried to explain in section 2.4.2 when writing:
"The variance is linked to measurement errors; i.e., a mix between the influence of the SNR and other random perturbations of the measure, such as variations in the physics of the problem. In the present problem, such perturbations are averaged when going from the time-space to the frequency-wavenumber

domains, and thus the variance reflects variations of frequency- wavenumber values around ground truth values."

Moreover, depending on the quality of the correlation function, these variations are not the same between all sets of dispersion curves. This is why we have chosen to estimate sigma from the simulated annealing part.

In an attempt to make this clearer in the manuscript, we have added the following explanations.

"However, $\sigma^2$ is data-dependent. In our approach, we are inverting dispersion curves, so it should account for uncertainties in the dispersion curves, not those in the measurements. Unfortunately, there is no linear relationship between both, because of the dispersion of the QS mode. Depending on the quality of the correlation function, these uncertainties are not the same between all sets of dispersion curves. Hence $\sigma^2$ cannot be set *a priori* for all of the inversions in this study, or else it would be a coarse approximation."

They subsequently use MCMC to obtain a posterior PDF with a uniform (limited) range prior distribution for each parameter, but do not show the covariances of the parameters. A posterior illustration of parameter covariances would also be helpful to illustrate the tradeoff space between the four parameters.

The range of prior distribution was chosen so as to include values found in literature for first year ice. We have added the following justification:

For first year ice, in situ measurements of density range from 840 to 910 kg/m3 for the ice above the waterline, and 900 to 940 kg/m3 for the ice below the waterline (Timco and Frederking, 1996). Poisson's ratio varies between 0.25 and 0.4, Young's modulus between 2 and 6 GPa (Anderson, 1958; Timco and Weeks, 2010). In particular, in the Van Mijen fjord, Romeyn et al. (2021) reported a Young's modulus around 2.5 GPa with a Poisson's ratio of 0.33, and Morozov et al (2012) reported a Young's modulus around 3 GPa and a Poisson's ratio of 0.3. (Moreau et al, 2014c), reported a Young's modulus around 4 GPa and a Poisson's ratio of 0.32 at Vallunden. The slightly higher value of Young's modulus at Vallunden, in comparison with those directly in the Van Mijen fjord, is likely attributable to the protected physical setting of the study site, and support from the surrounding shoreline at the moraine. Regarding the thickness, ice drillings indicated that it was systematically less than 1 m. Hence, we assume for the prior distributions that the model parameters have equal probability over a range of values that contains the above-referenced values.

We have added a new figure in Appendix B (see below), which shows the covariance between parameters. We have also modified the comments regarding the PDF accordingly:

Interestingly, the covariance of the parameters (see figure B1 in Appendix B) indicates that Poisson's ratio, despite being well-constrained, seems rather uncorrelated from the other parameters. On the other hand, Young's modulus, density and thickness appear to be strongly correlated, despite the density being not very well-constrained.

The observations seem to indicate that density having a flatter PDF reflects the limits of our forward model more than it is an indicator of the limits of the methodology. The model is only sensitive to the effective properties of the {ice+snow} system, because it cannot account for the snow layer (about 40 cm thick, on average), which modifies the effective properties. The weight of the snow layer modifies the density of the {ice+snow} system more than it does its rigidity (Young's modulus) and expansion/contraction (Poisson's ratio). Presumably, a forward model able to account for snow would be a significant improvement, which should constrain the density in a better way. The development of such a forward model is therefore an important follow up of this work.

[Figure]

Figure B1. Covariance between Young's modulus, thickness, density and Poisson's ratio, with the associated correlation coefficient

It is interesting that the inferred ice thicknesses differ systematically in the NS and EW directions in Figure 7. Can the authors elaborate on this (in line 270 they allude to special variations in the physics of the problem, but I believe what they meant to indicate was something like "spatial variations in ice model parameters".

We meant "spatial variations" indeed. This has been modified. We do not have a clear explanation for this. A likely explanation could be related to the elongated geometry of the moraine in the NS direction, which could have a mechanical effect to the ice while it is growing. Also, the channel to the north, where water current is present could have an effect. However, it is noteworthy that the differences between both directions are not statistically significant, since the uncertainties overlap, so it could also be linked to the quality of the Green's function retrieval. This remark was added to the manuscript.

All things considered, this is a significant work with a unique data set that could use a bit more polishing to realize its potential, and will no doubt be valuable for further analysis.